# Tight High-Probability Bounds for Nonconvex Heavy-Tailed Scenario under Weaker Assumptions

**Weixin An[1], Yuanyuan Liu[1],\*Fanhua Shang[2],\*Han Yu[3], Junkang Liu[2], Hongying Liu[4,5]\***

[1]Key Laboratory of Intelligent Perception and Image Understanding of Ministry of Education,
School of Artificial Intelligence, Xidian University, China
[2]School of Computer Science and Technology, Tianjin University, China
[3]College of Computing and Data Science, Nanyang Technological University, Singapore
[4]Medical School, Tianjin University, China
[5]Peng Cheng Lab, Shenzhen, China
`weixinanut@163.com, yyliu@xidian.edu.cn, fhshang@tju.edu.cn`
`han.yu@ntu.edu.sg, junkangliukk@gmail.com, hyliu2009@tju.edu.cn`

## Abstract

Gradient clipping is increasingly important in centralized learning (CL) and federated learning (FL). Many works focus on its optimization properties under strong assumptions involving Gaussian noise and standard smoothness. However, practical machine learning tasks often only satisfy weaker conditions, such as heavy-tailed noise and $(L_0, L_1)$-smoothness. To bridge this gap, we propose a high-probability analysis for clipped Stochastic Gradient Descent (SGD) under these weaker assumptions. Our findings show a better convergence rate than existing ones can be achieved, and our high-probability analysis does not rely on the bounded gradient assumption. Moreover, we extend our analysis to FL, where a gap remains between expected and high-probability convergence, which the naive clipped SGD can not bridge. Thus, we design a new Federated Clipped Batched Gradient (FedCBG) algorithm, and prove the convergence and generalization bounds with high probability for the first time. Our analysis reveals the trade-offs between the optimization and generalization performance. Extensive experiments demonstrate that FedCBG can generalize better to unseen client distributions than state-of-the-art baselines.

## 1 Introduction

Gradient clipping has proven effective in training vision and language models [53, 56]. Many studies demonstrated an optimal convergence rate of $\mathcal{O}(T^{-\frac{1}{2}})$ under a finite-variance assumption, where $T$ is the number of iterations or communication rounds. However, recent studies [54, 14] pointed out that assuming finite-variance noise is overly optimistic for modern machine learning tasks. Instead, it is more appropriate to assume that the noise has a bounded $p$-th moment, as stated in Assumption 3 below (**the first weaker assumption**), which is called heavy-tailed regime. This assumption brings significant challenges for theoretical analysis. Attempts to establish the convergence rate under this assumption have been made. For example, Zhang et al. [54] showed that clipped SGD achieves the state-of-the-art convergence rate in expectation. In practice, models are usually trained only once due to the long training process. Thus, Cutkosky and Mehta [7], Nguyen et al. [35], Puchkin et al. [37] studied high-probability convergence, offering a stronger guarantee for each individual run.

However, the above high-probability results are achievable only under standard smoothness. Works have demonstrated that some language and vision models [53, 46] can not satisfy the standard

---

*Corresponding authors

39th Conference on Neural Information Processing Systems (NeurIPS 2025).

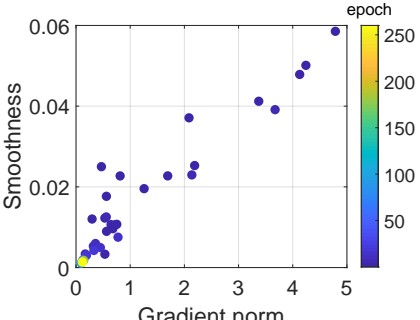
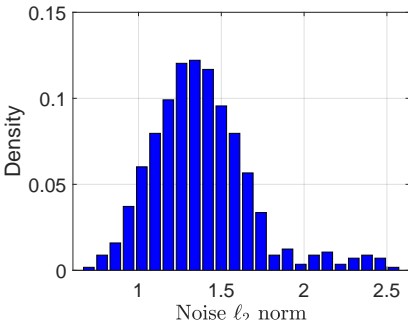

Figure 1: Gradient norm vs estimated Lipschitz smoothness (left) and gradient noise distribution (right) during training for AWD-LSTM [34] on the PTB dataset. Local smoothness positively correlates with gradient norm instead of a constant, which satisfies Assumption 2. The gradient noise exhibits a heavy-tailed behavior and its norm can be as large as 2.5. Similar phenomena also appeared in the Shakespeare dataset, as shown in Fig. 2.

smoothness assumption. Instead, Assumption 2 (**the second weaker assumption**) applies. Zhang et al. [53] first analyzed the convergence properties of clipped SGD under the $(L_0, L_1)$-smoothness assumption, which covers many large language models [6, 18].

The two types of works focus on the single weaker condition, but both conditions can appear in the same model, which can be verified in Fig. 1. Thus, there is an urgent need for analysis under both weaker conditions. Besides, these methods mentioned above focus on optimization, not including the generalization properties. Li and Liu [24, 25] first analyzed the generalization of clipped SGD, but both depend on the bounded gradient assumption, which is stronger than Assumption 3 and can not hold even when $f$ is quadratic.

As for federated learning (FL) [10, 40, 48, 38, 33, 39], heavy-tailed noise also exists. The works [45, 44, 47] focuses on this issue. The most related work to our paper is [47]. Their analysis is in expectation, which offers a weak guarantee for each single run. Besides, their analysis depends on the restrictive assumption that local gradients are bounded, which may not hold even for the quadratic function and the independent Gaussian random variables. In addition, they focus on the optimization performance under the standard smoothness assumption. In summary, there is a lack of studies jointly considering the optimization and generalization properties under weaker conditions in the FL setting.

The above analysis naturally raises the following questions:

*Q1: Can we analyze the clipped methods under only weaker conditions such as heavy-tailed noise and $(L_0, L_1)$-smoothness assumptions in high probability?*

*Q2: Can the analysis in CL inspire to design an FL algorithm to achieve the convergence rate matching the lower bound under weaker conditions?*

*Q3: Can we analyze the high-probability generalization properties under weaker assumptions for both CL and FL?*

## 1.1 Contributions

To answer these questions, we summarize our contributions as follows:

• By induction, we prove a faster convergence rate of the clipped SGD under the weaker conditions. By carefully choosing clipping parameter, we obtain a convergence rate $\mathcal{O}(T^{\frac{2-2p}{3p-2}} \log^{\frac{2p-2}{2p-1}} \frac{1}{\delta})$ with a probability of at least $1 - \delta$, $\delta \in (0, 1)$, which improves existing high-probability bound $\mathcal{O}(T^{\frac{2-2p}{3p-2}} \log^2 \frac{1}{\delta})$ ($p \in (1, 2]$). Interestingly, our analysis does not rely on the bounded gradient assumption used in [7, 24, 25]. Besides, we provide the generalization analysis for the first time.

• We design a new Federated Clipped Batched Gradient (FedCBG) algorithm for FL under weaker assumptions. We creatively prove the bounded variance of batch gradients, which opens the door to analyzing batch gradients under the heavy-tailed scenario. Then, we prove that FedCBG can achieve the advanced convergence rate of $\mathcal{O}((mKT)^{\frac{2-2p}{3p-2}} \log^{\frac{p-1}{p}} \frac{T}{\delta})$, where $m$ and $K$ is the number

of clients and local iterations, respectively. Finally, we provide a generalization upper bound for the federated setting for the first time. Our analysis reveals the trade-off between optimization and generalization. A summary of our theoretical contributions can be found in Tables 1 and 2 below.

## 2 Related Work

### 2.1 Existing analysis under weaker conditions

Many works such as [54, 47] have shown that there exists heavy-tailed noise in many applications. Analyzing convergence under this scenario is more challenging than for light-tailed noise (e.g., Gaussian noise) due to the unbounded variance, which makes most existing proof techniques inapplicable. The first type of analysis addressed this issue by assuming the bounded gradient $\mathbb{E}_t[\|\nabla f(x_t; \xi_{j_t})\|^p] \leq G^p$ [54, 47, 25]. However, this assumption is strong and cannot hold even when the loss is quadratic. As a comparison, our inductive analysis only needs the weakened Assumption 3, which is also used in works [35, 30]. Besides, the loss functions of many language and vision models can not satisfy the standard smoothness but rather a weaker $(L_0, L_1)$-smoothness [53, 23]. $(L_0, L_1)$-smoothness assumption is applied by the works [53, 51, 6, 18, 23] under the light-tailed noise. As for the heavy-tailed noise, the existing works [54, 29] only analyzed the expected rather than high-probability convergence rates as shown in Table 4 in the Appendix. In contrast, we propose tight high-probability analysis under both weaker conditions, offering a stronger guarantee for each individual training.

### 2.2 Optimization properties for clipped methods

In centralized learning settings, existing studies [54, 29] focused on clipping for the heavy-tailed scenario and analyzed the convergence bounds in expectation. Besides, works such as [24, 25, 35, 30] provide high-probability analysis, which matches the lower bound in expectation. However, the order of $\log \frac{T}{\delta}$ is at least 2 as shown in Table 1. We reduce this order to $\frac{2p-2}{2p-1}$ as shown in the same table. In FL, to the best of our knowledge, there is only one work [47] analyzing convergence rates under the heavy-tailed noise. However, they focus on expected rather than high-probability analysis, and optimization properties rather than generalization aspects.

### 2.3 Generalization for nonconvex problems

Existing generalization analysis contains three types: 1) in expectation [5, 9, 15, 21, 22, 36, 43, 50], 2) high probability [12, 32, 13, 20, 16], and 3) information theory [1, 4, 52]. For example, Hardt et al. [15] pioneered generalization analysis in expectation based on stability. However, their analysis requires a very small step size, which leads to an exponential number of iterations. As a comparison, the high-probability analysis allows the constant step size, which controls the generalization error and makes the optimization error decay faster. Besides, it can provide a stronger guarantee for each single run and is a tighter criterion for bounded losses [1]. As for the information-theoretic analysis, they are usually algorithm-independent [3]. In this paper, we provide the high-probability upper bounds for optimization and generalization and focus on their joint perspective.

## 3 Preliminaries

**Problem setting:** In this paper, we focus on the clipped methods for solving the problems in both CL and FL settings. For the CL setting, the population loss is defined as:

$$F(x) = \mathbb{E}_{\xi \sim P_\xi} f(x; \xi), \tag{1}$$

where the loss function $f$ is nonconvex w.r.t. $x$, $x$ is network weights, and one sample $\xi$ is sampled from the distribution $P_\xi$. Normally, the population risk $F(x)$ is used for generalization but is computationally invisible and it can only be estimated using the empirical risk $F_S(x) := \frac{1}{n} \sum_{i=1}^{n} f(x; \xi_i)$.

FL [31] allows multiple participants to share model training results but not data, reducing the risk of data leakage. FL usually addresses the following problem:

$$F(x) := \mathbb{E}_{i \sim \mathcal{P}} \{ F_i(x) := \mathbb{E}_{\xi_j \sim P_i} f(x; \xi_j) \}, \tag{2}$$

where $f(x; \xi_j)$ is the loss at sample $\xi_j$, $\xi_j$ is sampled from the local distribution $P_i$, each client $i$ is sampled from a meta-distribution $\mathcal{P}$. We define the client empirical risk by $f_i(x) := \frac{1}{n_i} \sum_{j=1}^{n_i} f_i(x; \xi_j)$

and the empirical risk on the participating training client data is defined by $F_S(x) := \frac{1}{m}\sum_{i=1}^m f_i(x)$, where $n_i$ is the number of samples of the $i$-th client and $m$ is the number of participating clients. Generalization research in FL includes the performance gap on unseen client data and unseen client distributions. For the former, the CL can provide help. In this paper, we focus on the latter. For example, Problem (2) is common in the cross-device FL setting, where $m$ is generally large and it is reasonable to sample from a meta-distribution to model local distributions of clients [49], which makes it clear the generalization to non-participating clients.

**Notation:** We use lower-case letters to denote vectors. For a differentiable function $f$, $\nabla f(x)$ is the gradient of $f$ at $x$. We let $\mathcal{F}_t$ be the natural filtration for the algorithms. $\mathbb{E}_t$ is used to denote $\mathbb{E}[\cdot|\mathcal{F}_{t-1}]$ for brevity.

**Assumption 1** (Bounded Function). *$F$ admits a finite lower bound, i.e., $F^* = \inf_x F(x) > -\infty$.*

**Assumption 2** (($L_0, L_1$)-Smoothness). *The smoothness of the function $F_S$ means that for $\forall x, y$ satisfying $\|x - y\| \leq \frac{1}{L_1}$, $\|\nabla F_S(x) - \nabla F_S(y)\| \leq (L_0 + L_1\|\nabla F_S(x)\|)\|x - y\|$ holding with smoothness parameter $\ell = L_0 + L_1\|\nabla F_S(x)\|$.*

*For the federated setting, the local ($L_0, L_1$)-Lipschitz continuous gradient for each client means $\|\nabla f_i(x) - \nabla f_i(y)\| \leq (L_0 + L_1\|\nabla f_i(x)\|)\|x - y\|$. When $L_1 = 0$, they become standard smoothness.*

**Assumption 3** (Heavy-tailed Noise). *For the centralized setting, the stochastic gradient estimator is unbiased, i.e., $\mathbb{E}[\nabla f(x; \xi)] = \nabla F_S(x)$. Besides, the gradient noise satisfies the heavy-tailed condition $\mathbb{E}_\xi\|\nabla F_S(x) - \nabla f(x; \xi)\|^p \leq \sigma^p, p \in (1, 2]$.*

*For the federated setting, the local gradient estimator is unbiased, i.e., $\mathbb{E}[\nabla f_i(x; \xi)] = \nabla f_i(x)$. Besides, the local stochastic gradient noise in the $i$-th client follows the heavy-tailed distribution, i.e., $\mathbb{E}_\xi\|\nabla f_i(x) - \nabla f_i(x; \xi)\|^p \leq \sigma^p, p \in (1, 2]$.*

Many works such as image classification [42], training the large language models [54] and FL [47] have shown that stochastic gradient noise usually follows the heavy-tailed distribution, which is also corroborated by Fig. 1. Some works [7, 25] have made this assumption concrete to that the stochastic gradients are bounded in $p$-th moment, i.e., $\mathbb{E}_t\|\nabla f(x_t, \xi_{j_t})\|^p \leq G^p$ (or $\mathbb{E}_t\|\nabla f_i(x_t, \xi_{j_t})\|^p \leq G^p$ in FL), for some $G > 0$. However, it does not hold even when $f$ is quadratic and $\nabla f(x; \xi) - \nabla F_S(x)$ (or $\nabla f_i(x; \xi) - \nabla f_i(x)$ in FL) is an independent centered Gaussian random variable. In contrast, Assumption 3 is weaker. In this paper, we focus on the analysis under Assumptions 1-3.

# 4 Tighter High-probability Bounds in the Centralized Setting

To answer Q1, we first consider the optimization properties of clipped SGD in Subsection 4.1. Besides, we prove its generalization bound in Subsection 4.2.

## 4.1 Tighter high-probability convergence under weaker conditions

The pseudocode of clipped SGD is shown in Algorithm 1. In each iteration, clipped SGD performs gradient descent along the clipped gradient $\widetilde{\nabla} f(x_t; \xi_{j_t})$.

We extend the analysis in [35] to the ($L_0, L_1$)-smoothness assumption, which can cover more applications. In Theorem 1, we propose better parameter choices and prove a faster convergence rate than existing analyses such as [35, 30].

---
**Algorithm 1** Clipped SGD

**Initialize:** $x_0$, step size $\eta$ and clipping parameter $\lambda$.
 1: **for** $t = 0, 1, \ldots, T - 1$ **do**
 2:     Draw i.i.d. $\xi_{j_t}$ stochastic sample;
 3:     $\widetilde{\nabla} f(x_t; \xi_{j_t}) = \min\{1, \frac{\lambda}{\|\widetilde{\nabla} f(x_t; \xi_{j_t})\|}\}\nabla f(x_t; \xi_{j_t})$;
 4:     $x_{t+1} = x_t - \eta\widetilde{\nabla} f(x_t; \xi_{j_t})$;
 5: **end for**
 6: Randomly draw $\hat{x}$ from $x_1, \ldots, x_T$ at uniform;
**Output:** $\hat{x}$.

---

**Theorem 1.** *We assume that Assumptions 1, 2 and 3 hold. If we choose $\lambda$ and $\eta$ satisfying $\lambda = \mathcal{O}(T^{\frac{1}{3p-2}}(\log\frac{T}{\delta})^{\frac{1}{1-2p}}), \eta = \mathcal{O}(T^{\frac{-p}{3p-2}}(\log\frac{T}{\delta})^{\frac{2-2p}{2p-1}})$, where $\mathcal{R} = L_0 + 2(L_1 + 1)R$, the constant $R \geq 4\Delta_1 L_1 + 4\sqrt{\Delta_1^2 L_1^2 + L_0\Delta_1}$, $\Delta_1 = F_S(x_1) - F^*$, $\rho = \max\{\log\frac{4T}{\delta}, 1\}$, the clipped SGD (Algorithm 1) can achieve the convergence rate of $\frac{1}{T}\sum_{t=1}^T\|\nabla F_S(x_t)\|^2 = \mathcal{O}(T^{\frac{2-2p}{3p-2}}(\log\frac{T}{\delta})^{\frac{2p-2}{2p-1}})$ with the probability at least $1 - \delta$ for any $\delta \in (0, 1)$.*

Theorem 1 offers a new high-probability optimization bound for clipped SGD. According to Jensen's inequality, the bound implies that $\frac{1}{T}\sum_{t=1}^{T}\|\nabla F_S(x_t)\| \leq \mathcal{O}(\log^{\frac{p-1}{2p-1}}\frac{T}{\delta}/T^{\frac{p-1}{3p-2}})$, which matches the lower bound $\Omega(T^{\frac{p-1}{3p-2}})$ in [54] up to a logarithmic factor. Compared with existing results, our bound has the following advantages.

Table 1: Comparison of existing high-probability (HP) analysis in centralized learning (CL). We use $\frac{1}{T}\sum_{t=1}^{T}\|\nabla F_S(x_t)\|^2$ and $\frac{1}{T}\sum_{t=1}^{T}\|\nabla F(x_t)\|^2$ as the criterion in high probability. Abbreviation: Standard smoothness (SS), $(L_0, L_1)$-smoothness $((L_0, L_1))$, Heavy-tailed (HT), Theorem (Th.). Here, $G$ is a constant, $\delta \in (0, 1)$, and $p \in (1, 2]$. It can be seen that our Theorems 1 and 2 achieve better optimization convergence and the state-of-the-art generalization bound under weaker assumptions.

| Methods | Assumptions | | Additional Assumptions | Bounds | |
| | Smooth | Noise | | Optimization | Generalization |
| --- | --- | --- | --- | --- | --- |
| [24] | SS | HT | $\eta\|\nabla F_S(x_t)\| \leq G$ | $\mathcal{O}(\frac{\log T}{T^{1/2}}\log^2\frac{1}{\delta})$ | $\mathcal{O}((\frac{d}{n})^{\frac{p-1}{3p-2}}\log^{\frac{2p-2}{2p-1}}(\sqrt{\frac{n}{\delta^2 d}}))$ |
| [25] | SS | HT | $\mathbb{E}_t[\|\nabla f(x_t; \xi_{j_t})\|^p] \leq G^p$ | $\mathcal{O}(T^{\frac{2-2p}{3p-2}}\log\frac{1}{\delta})$ | $\mathcal{O}((\frac{d}{n})^{\frac{p-1}{3p-2}}\log^{\frac{2p-2}{2p-1}}(\sqrt{\frac{n}{\delta^2 d}}))$ |
| [35] | SS | HT | —— | $\mathcal{O}(T^{\frac{2-2p}{3p-2}}\log^{\frac{p}{p-1}}\frac{T}{\delta})$ | —— |
| [30] | SS | HT | —— | $\mathcal{O}(T^{\frac{2-2p}{3p-2}}\log^2\frac{T}{\delta})$ | —— |
| Th. 1, 2 | $(L_0, L_1)$ | HT | —— | $\mathcal{O}(T^{\frac{2-2p}{3p-2}}\log^{\frac{2p-2}{2p-1}}\frac{T}{\delta})$ | $\mathcal{O}((\frac{d}{n})^{\frac{p-1}{3p-2}}\log^{\frac{2p-2}{2p-1}}(\sqrt{\frac{n}{\delta^2 d}}))$ |

• **Addressed the challenges under the weaker assumption.** In our analysis, the clipped SGD can deal with the $(L_0, L_1)$-smoothness rather than only standard smoothness. The generalized smoothness increased analysis difficulty due to extra $\|\nabla F_S(x)\|$ in the upper bound ($(L_0, L_1)$-smoothness makes the gradient upper bound implicit in an inequality, which complicates the analysis). Specifically, it can lead to a high-order term containing $\|\nabla F_S(x)\|$. The previous works like [51] keep it till the end and use the boundedness of clipped gradients to choose step size $\eta$. Instead, Zhang et al. [53] chooses carefully clipped step size $\min\{\eta, \frac{\eta\lambda}{\|\nabla F_S(x)\|}\}$ to achieve convergence. But they focus on noise with bounded variance or need additional assumption $\|\nabla f(x; \xi) - \nabla F_S(x)\| \leq \sigma$, which are not practical even when the loss is quadratic. However, the noise variance can not be easily bounded under the heavy-tailed scenario, and the boundedness of clipped gradients and clipped step sizes can not be used for high-order terms. Thus, this paper still faces the challenge of high-order terms.

Inspired by the analysis in [11, 23] for bounded variance, we prove Theorem 1 by induction. In Appendix B.1, we show how to use induction arguments to handle $(L_0, L_1)$-smoothness and remove the bounded gradient assumption, and here we give a proof sketch.

**Proof sketch.** The key in the convergence rate analysis is to show that $\|\nabla F_S(x_t)\| \leq \frac{\lambda}{2}$. By induction hypothesis at $l$ ($l \leq t$), we creatively solve a quadratic inequality w.r.t. $\|\nabla F_S(x)\|$ so that the gradient $\|\nabla F_S(x)\|$ can be controlled under the $(L_0, L_1)$-smoothness when $\lambda$ is greater than a constant. Based on these, we construct a new martingale difference sequence $\sum_{t=0}^{l-1}(L_1\eta^2\|\nabla F_S(x_t)\| - \eta)\langle\nabla F_S(x_t), \theta_t^a\rangle$ produced by $(L_0, L_1)$-smoothness, which does not appear in standard analysis like in [35], where $\theta_t^a = \widetilde{\nabla}f(x_t; \xi_{j_t}) - \mathbb{E}_t[\widetilde{\nabla}f(x_t; \xi_{j_t})]$. Next, by carefully choosing $\lambda$ and $\eta$, we can obtain the following induction $\Delta_{T+1} + \frac{\eta}{4}\sum_{t=1}^{T}\|\nabla F_S(x_t)\|^2 \leq 2\Delta_1$ with the probability at least $1 - \delta$, thereby achieving the desired convergence rate.

• **Tighter convergence bound.** We analyzed the parameter selection in [35] and found that $\mathcal{O}(T^{\frac{2-2p}{3p-2}})$ is already tight but the order of $\log\frac{T}{\delta}$ can be reduced. By analyzing the inequalities that $\lambda$ satisfies in our induction, we set $\lambda = \mathcal{O}(T^{\frac{1}{3p-2}}(\log\frac{T}{\delta})^{\frac{1}{1-2p}})$ and $\eta = \mathcal{O}(T^{\frac{-p}{3p-2}}(\log\frac{T}{\delta})^{\frac{2-2p}{2p-1}})$, which yields a tighter convergence rate on the logarithmic factor compared with [35, 30], as shown in Table 1.

### 4.2 Generalization bound under weaker assumptions

In addition to optimization, we analyze the generalization bound of clipped SGD to answer Q3. We use the term $\|\nabla F(x_t)\|^2$ to estimate this bound. Similar criteria can be found in [25, 20].

**Theorem 2.** *We assume that Assumptions 1, 2 and 3 hold. We set the same step size and clipping parameter as Theorem 1. If we choose $T = \mathcal{O}\left(\sqrt{\frac{n}{d}}\right)$, then with probability at least $1 - \delta$, Algorithm 1 can achieve $\frac{1}{T}\sum_{t=1}^{T}\|\nabla F(x_t)\|^2 \leq \mathcal{O}((\frac{d}{n})^{\frac{p-1}{3p-2}}\log^{\frac{2p-2}{2p-1}}(\frac{1}{\delta}\sqrt{\frac{n}{d}}))$.*

Theorem 2 shows that clipped SGD can guarantee the generalization bound of the order $\mathcal{O}((\frac{d}{n})^{\frac{p-1}{3p-2}}\log^{\frac{2p-2}{2p-1}}(\frac{1}{\delta}\sqrt{\frac{n}{d}}))$ under weaker assumptions, such as heavy-tailed noise and $(L_0, L_1)$-smoothness. Besides, Theorem 2 is the first high-probability generalization analysis without bounded gradient assumption. For clarification, we provide a proof sketch.

**Proof sketch.** We estimate the term $\|\nabla F(x_t)\|^2$ as follows $\|\nabla F(x_t)\|^2 \leq 2\|\nabla F_S(x_t)\|^2 + 2\|\nabla F(x_t) - \nabla F_S(x_t)\|^2$. The first term is optimization error and we can bound it by Theorem 1. The second term is generalization error due to approximating the true gradient with its empirical counterpart and we bound it by generalized uniform convergence as shown in Lemma 5. In Lemma 5, the value of $R$ needs to be quantified. We prove that the generalization error increases as training progresses and we can choose $R = \max_{1 \leq t \leq T}\|x_t\|$. Next, we decompose $\|x_t\|$ into $A_1, A_2, A_3$ by Triangle Inequality and combine them with our inductive Lemma 6 to get $\|x_{t+1}\| = \mathcal{O}(T^{\frac{2p-1}{3p-2}}/\log^{\frac{p-1}{2p-1}}\frac{T}{\delta})$. Along this line of thought, we successfully bounded $\|\nabla F(x_t)\|^2$.

**Advantages compared with existing analysis**

• We offer generalized uniform convergence for $(L_0, L_1)$-smooth objective as shown in Lemma 5, which generalizes the results in [20, 25].

• Remove the bounded gradient assumption. We use our induction Lemma 6, which can guarantee that $\|\nabla F_S(x_t)\| \leq \frac{\lambda}{2}$ with the probability at least $1 - \delta$. This analysis removes the bounded gradient assumption in [25], i.e., $\mathbb{E}_t[\|\nabla F_S(x_t; \xi_t)\|^p] \leq G^p$ and achieves the first generalization upper bound.

## 5 The Proposed `FedCBG` Algorithm for the Federated Setting

As we discussed above, heavy-tailed noise also exists in FL. To answer Q2, we extend Theorems 1 and 2 to FL, which inspires us to design an `FedCBG` algorithm to match the optimization lower bound. Besides, we provide the high-probability generalization bound for the first time.

### 5.1 Federated clipped batch gradient algorithm

To handle the heavy-tailed noise, we design a Federated Clipped Batch Gradient (`FedCBG`) algorithm as shown in Algorithm 2, which mainly contains two parts:

• In the client, we use clipped batch gradient $\widetilde{\nabla}f_i(x_{t,i}^k; \boldsymbol{\xi}_{t,i}^k)$ to perform gradient descent, where $\widetilde{\nabla}f_i(x_{t,i}^k; \boldsymbol{\xi}_{t,i}^k) = \min\{1, \frac{\lambda}{\|\nabla f_i(x_{t,i}^k; \boldsymbol{\xi}_{t,i}^k)\|}\}\nabla f_i(x_{t,i}^k; \boldsymbol{\xi}_{t,i}^k)$ and $\boldsymbol{\xi}_{t,i}^k = \{(\xi_{t,i}^k)_j\}_{j=1}^b$, which is different from the existing methods such as [47], where they use a single sample in each local update. This difference is one of the key reasons why we obtain a convergence rate matching the lower bound under the more difficult criterion, i.e., in high probability.

• In the server, we design a "sum-aggregation" paradigm $x_{t+1} = x_t - \gamma\sum_{i=1}^m \widetilde{\Delta}_t$, which is the other reason why our `FedCBG` can achieve the convergence rate in high probability matching the lower bound.

---

**Algorithm 2** `FedCBG` Algorithm

---

**Initialize:** Initial point $x_0$, local step size $\eta$, global learning rate $\gamma$ and clipping parameter $\lambda$.

1: **for** $t = 0, 1, \ldots, T - 1$ (communication round) **do**
2:     **for** each client $i \in [m]$ in parallel **do**
3:         Update local model: $x_{t,i}^0 = x_t$.
4:         **for** $k = 0, \cdots, K - 1$ (local update step) **do**
5:             Draw i.i.d. stochastic samples $\boldsymbol{\xi}_{t,i}^k$;
6:             $x_{t,i}^{k+1} = x_{t,i}^k - \eta\widetilde{\nabla}f_i(x_{t,i}^k; \boldsymbol{\xi}_{t,i}^k)$;
7:         **end for**
8:         Send $\widetilde{\Delta}_t^i = \sum_{k=0}^{K-1}\widetilde{\nabla}f_i(x_{t,i}^k; \boldsymbol{\xi}_{t,i}^k)$ to the server.
9:     **end for**
10:    Global sum-aggregation at server:
11:       Server update: $x_{t+1} = x_t - \gamma\sum_{i=1}^m \widetilde{\Delta}_t$;
12:       Broadcasting $x_{t+1}$ to clients.
13: **end for**

**Output:** $x_T$.

---

## 5.2 Convergence rate of our `FedCBG` algorithm

To prove the convergence rate of `FedCBG`, we need to bound the variance of the batch gradient under the heavy-tailed noise assumption. Compared to the Gaussian noise (light-tailed) assumption, such analysis is more difficult. Fortunately, by Hölder Inequality and Markov's Inequality, we have proved an upper bound on this variance for the first time in Lemma 1.

**Lemma 1** (**Batch gradient variance bound**). *If Assumptions 3 holds and $\|\nabla f_i(x_t)\| \leq \frac{\lambda}{2}$, $\forall i \in [m]$, for batch gradient $\nabla f_i(x_{t,i}^k; \boldsymbol{\xi}_{t,i}^k) = \frac{1}{b} \sum_{\xi_{t,i}^k \in \boldsymbol{\xi}_{t,i}^k} \nabla f_i(x_{t,i}^k; \xi_{t,i}^k)$, we have the batch gradient variance bound*

$$\mathbb{E}_t[\|\nabla f_i(x_{t,i}^k) - \nabla f_i(x_{t,i}^k; \boldsymbol{\xi}_{t,i}^k)\|^2] \leq \frac{3\sigma^p \lambda^{2-p}}{b}. \tag{3}$$

**Remark 1.** *In Lemma 1, we provide the first upper bound for batched gradient variance under the heavy-tailed noise. In our high-probability analysis, $b > 1$ provides one parameter of freedom for choosing the clipping parameter $\lambda$, thereby achieving the convergence rate matching the lower bound in expectation. Specifically, we set $b = \mathcal{O}((mKT)^{\frac{2p-2}{3(3p-2)}})$, which allows us to choose $\lambda = \mathcal{O}((mKT)^{\frac{1}{2(3p-2)}})$, thereby achieving the desired convergence rate.*

Inspired by the definitions of $\theta_t^a$ and $\theta_t^b$ ($\theta_t^b = \mathbb{E}_t[\widetilde{\nabla} f(x_t; \xi_{j_t})] - \nabla F_S(x_t)$) in the centralized setting, we construct three errors in the federated setting: stochastic batch error $\epsilon_t$, the clipped batch gradient deviation $\epsilon_t^a$, and the bias $\epsilon_t^b$ between the expected clipped batch gradient and full gradient, where $\epsilon_t^a = \frac{1}{mK} \sum_{i=1}^m \sum_{k=1}^K (\widetilde{\nabla} f_i(x_{t,i}^k; \boldsymbol{\xi}_{t,i}^k) - \mathbb{E}_t[\widetilde{\nabla} f_i(x_{t,i}^k; \boldsymbol{\xi}_{t,i}^k)])$, $\epsilon_t^b = \frac{1}{mK} \sum_{i=1}^m \sum_{k=1}^K \mathbb{E}_t[\widetilde{\nabla} f_i(x_{t,i}^k; \boldsymbol{\xi}_{t,i}^k)] - \nabla F_S(x_t)$, and $\epsilon_t = \epsilon_t^a + \epsilon_t^b$. Based on Lemma 1, we analyze their upper bounds in Lemma 2.

**Lemma 2.** *For Algorithm 2, $\forall t \in [T]$, we have $\|\epsilon_t^a\| \leq 2\lambda$. Besides, if $\|\nabla f_i(x_t)\| \leq \frac{\lambda}{2}$, there is $\|\epsilon_t^b\| \leq \frac{12\sigma^p \lambda^{1-p}}{b}$ and $\mathbb{E}_t[\|\epsilon_t^a\|^2] \leq \frac{100\sigma^p \lambda^{2-p}}{mKb}$.*

Lemma 2 shows that the clipped batch gradient can add a parameter of freedom $b$ compared with [35, 37], which relaxes the conditions for choosing hyperparameters. Now, we begin to prove the convergence rate of our `FedCBG` algorithm. The key to our derivation lies in Lemma 3.

**Lemma 3.** *For $1 \leq N \leq T+1$, let $E_N'$ be the event that for all $l = 1, \cdots, N$,*

$$\Delta_l' + \frac{\gamma mK}{2} \sum_{t=1}^{l-1} \|\nabla F_S(x_t)\|^2 \leq \Delta_1' + \gamma mK \sum_{t=1}^{l-1} [(1 + L_1 \|\nabla F_S(x_t)\|)(\|\epsilon_t^a\|^2 - \mathbb{E}_t[\|\epsilon_t^a\|^2])$$

$$+ L_1 \|\nabla F_S(x_t)\| (\langle \epsilon_t^a, \nabla F_S(x_t) \rangle + \|\nabla F_S(x_t)\| \|\epsilon_t^b\|)] + \frac{L_1 \gamma^2 m^2 K^2}{2} \sum_{t=1}^{l-1} \|\nabla F_S(x_t)\|^3 \tag{4}$$

$$+ \gamma mK \sum_{t=1}^{l-1} (1 + L_1 \|\nabla F_S(x_t)\|)(\|\epsilon_t^b\|^2 + \mathbb{E}_t[\|\epsilon_t^a\|^2]) \leq 2\Delta_1'.$$

*Then $E_N'$ happens with probability at least $1 - \frac{(N-1)\delta}{T}$ for each $N \in [T]$.*

Lemma 3 explains why our Algorithm 2 can achieve a convergence rate matching the lower bound. Specifically, in our inductive analysis, we focus on constructing the martingale difference sequences $\{\|\epsilon_t^a\|^2 - \mathbb{E}_t[\|\epsilon_t^a\|^2]\}$ and $\{\langle \epsilon_t^a, \nabla F_S(x_t) \rangle\}$ and bound them in high probability by Freedman's inequality. Besides, by induction hypothesis at $l$ ($l \leq t$), we also creatively solve a quadratic inequality w.r.t. $\|\nabla F_S(x)\|$ and $\|\nabla f_i(x)\|$ so that they can be controlled under the $(L_0, L_1)$-smoothness when $\lambda$ is greater than a constant. Then, we leverage Lemma 2 and choose appropriate $\eta, \gamma, b$ and $\lambda$ to balance all the terms to achieve the desired convergence rate. Based on Lemma 3, we prove the convergence rate of Algorithm 2 as shown in Theorem 3.

**Theorem 3.** *We assume that Assumptions 1, 2 and 3 hold. If we choose $b = \mathcal{O}((mKT)^{\frac{2p-2}{3(3p-2)}})$, $\lambda = \mathcal{O}((mKT)^{\frac{1}{3(3p-2)}}/\rho^{\frac{1}{2p}})$, $\gamma = \mathcal{O}((mKT)^{\frac{-p}{3p-2}}/\rho^{\frac{p-1}{p}})$, $\eta = \mathcal{O}(\frac{\log^{\frac{1}{p}} \frac{T}{\delta}}{K^{\frac{17p-8}{4(3p-2)}}(mT)^{\frac{5p}{4(3p-2)}}})$, where $\mathcal{R}' = 1 + 2(\frac{L_1}{L_0} + 1)R'$, $R' \geq 4\Delta_1' L_1 + 4\sqrt{(\Delta_1')^2 L_1^2 + L_0 \Delta_1'}$, $\rho = \max\{\log \frac{4T}{\delta}, 1\}$, $\Delta_t' = F_S(x_t) - F^*$, and $\beta = \min\{\frac{32\mathcal{R}\rho}{L_0}, \frac{3}{4}, \frac{3L_0}{8L_1 R'}\}$, Algorithm 2 can achieve the convergence rate*

*of $\frac{1}{T}\sum_{t=1}^{T}\|\nabla F_S(x_t)\|^2 = \mathcal{O}((mKT)^{\frac{2-2p}{3p-2}}\log^{\frac{p-1}{p}}\frac{T}{\delta})$ with the probability at least $1-\delta$ for any $\delta \in (0,1)$.*

Table 2: Comparison of the existing analysis in FL. "AA" indicates whether the additional gradient boundedness assumption $\mathbb{E}_t[\|\nabla f_i(x_t;\xi_{j_t})\|^p] \leq G^p$ are required, and "LB" refers to the lower bound.

| Methods | Assumptions | | Criteria | AA | Bounds | |
| | Smooth. | Noise | | | Optimization | Generalization |
| --- | --- | --- | --- | --- | --- | --- |
| [47] | SS | HT | Exp | ✓ | $\mathcal{O}((mT)^{\frac{2-2p}{3p-2}}K^{\frac{4-2p}{3p-2}})$ | —— |
| [47] | SS | HT | Exp | ✓ | $\mathcal{O}((mKT)^{\frac{2-2p}{3p-2}})$ | —— |
| LB | SS | HT | Exp | ✗ | $\Omega((mKT)^{\frac{2-2p}{3p-2}})$ | —— |
| Th. 3, 4 | $(L_0,L_1)$ | HT | HP | ✗ | $\mathcal{O}((mKT)^{\frac{2-2p}{3p-2}}\log^{\frac{p-1}{p}}\frac{1}{\delta})$ | $\mathcal{O}\big((\frac{d}{n})^{\frac{p-1}{7p-6}}\log(\frac{1}{\delta}(\frac{n}{d})^{\frac{3p-2}{2(7p-6)}})\big)$ |

Theorem 3 shows that our `FedCBG` algorithm can achieve the desired convergence rate for the heavy-tailed noise setting. The size of $b$ is consistent with our intuition that the smaller $p$ is, the more sensitive the algorithm is to noise, the gradient differences between different samples may be large, thus a small $b$ can achieve an ideal convergence rate. This convergence rate $\mathcal{O}((mKT)^{\frac{2-2p}{3p-2}}\log^{\frac{p-1}{p}}\frac{T}{\delta})$ matches the lower bound proposed in [47] up to a logarithmic factor as shown in Table 2. Thus, `FedCBG` effectively reduces the number of communication rounds. Besides, our clipping parameter $\lambda$ is smaller than that of [47]. Small $\lambda$ is typically used and often leads to good performance [34, 55], as stated in [17]. The specific parameter choices and the inequalities they need to satisfy can be found in Appendix C.1. Compared with the state-of-the-art methods in [47], our analysis has the following advantages. **1)** Our analysis is in high probability and provides a stronger guarantee for a single run. **2)** Our analysis use the weaker Assumption 3 rather than the bounded gradient assumption, i.e., $\mathbb{E}_t[\|\nabla f_i(x_t;\xi_{j_t})\|^p] \leq G^p$. **3)** Our analysis use the weaker Assumption 2 rather than standard smoothness. Thus, our analysis can apply to a wider range of applications than existing methods.

### 5.2.1 Challenges and techniques for our analysis

In our analysis, we attempt to extend our Theorem 1 to the federated setting, but we find it is very difficult or even impossible to match the lower bound. The reasons are the following: a faster convergence rate requires a larger step size, but the inductive property requires a smaller step size. Thus, a contradiction arises. We balance the contradiction by addressing the following challenges.

**Construct high-probability criteria.** Starting from the smoothness of the function $F_S(x)$, we use our proposed "sum-aggregation" paradigm and $-\langle a,b\rangle = \frac{1}{2}\|a-b\|^2 - \frac{1}{2}\|a\|^2 - \frac{1}{2}\|b\|^2$ to handle the tricky inner product term $\langle \nabla F_S(x_t), x_{t+1} - x_t\rangle$. It helps to produce the term $\frac{\gamma mK}{2}\|\nabla F_S(x_t)\|^2$ and construct martingale difference sequences in Lemma 9, which constructs the high-probability criteria and relaxes the restrictions on parameter selection in the induction.

**Difficulty of the analysis in high probability.** Many upper bounds in expectation are usually tighter and more concise than those of high probability. For example, there is the bound $\mathbb{E}_t\|\theta_t^a\|^2 \leq 10\sigma^p\lambda^{2-p}$ but only the bound $\|\theta_t^a\|^2 \leq 4\lambda^2$ in the centralized setting, where $\lambda$ is usually of the order $\mathcal{O}(T^\alpha)$, $\alpha > 0$. A similar phenomenon also appears in federated learning, which makes the analysis difficult. Fortunately, we prove that our clipped batch gradient can provide one extra parameter of freedom to handle these rough upper bounds in Lemma 1.

**Weaker assumptions.** In FL, the difficulties caused by heavy-tailed noise and $(L_0,L_1)$-smoothness were addressed by induction and our martingale difference sequence, just like in CL.

### 5.3 Generalization bound for our `FedCBG` algorithm

In FL, there is no work jointly considering the optimization and generalization. To answer Q3, we analyze the generalization upper bound for our `FedCBG` in Theorem 4.

**Theorem 4.** *We assume that Assumptions 1, 2 and 3 hold. We choose the same parameter setting as in Theorem 3. If we choose $T = \mathcal{O}\big(\big(\frac{n}{d}\big)^{\frac{3p-2}{2(7p-6)}}/m^{\frac{6p-5}{7p-6}}\big)$ and $K = \mathcal{O}\big(\big(\frac{n}{d}\big)^{\frac{3p-2}{2(6p-5)}}\big)$, then with*

*probability at least $1 - \delta$, Algorithm 2 can achieve the generalization bound $\frac{1}{T}\sum_{t=1}^{T}\|\nabla F(x_t)\|^2 = \mathcal{O}\big((\frac{d}{n})^{\frac{p-1}{7p-6}}\log(\frac{1}{\delta}(\frac{n}{d})^{\frac{3p-2}{2(7p-6)}})\big)$.*

To the best of our knowledge, this generalization bound is the first upper bound for FL with heavy-tailed noise. If we set a small global learning rate $\gamma$ to obtain better generalization, the convergence speed will be slower, which reflects the trade-off between optimization and generalization.

## 6  Experiments

In this section, we evaluate our `FedCBG` algorithm against only FL algorithms FAT-clipping-PR (PR) and FAT-clipping-PI (PI) [47] that can theoretically handle heavy-tailed noise. We also compare the well-known FedAvg algorithm [31]. We test these methods on the CIFAR-10, CIFAR-100 [19] and Shakespeare [41] datasets. By the way, our goal is to compare the relative performance of `FedCBG` and baselines and larger models can achieve better performance on these datasets. All the experiments were performed on the GeForce RTX 2080Ti platform with the PyTorch framework.

**Training an LSTM only satisfying the weaker Assumptions 2 and 3.** Firstly, to verify that the federated scenarios may only meet weaker assumptions (i.e., Assumptions 2 and 3), we evaluate the smoothness and gradient noise distribution of a stacked LSTM as in [26] training on the Shakespeare dataset. We show smoothness and the histograms of gradient noise probability density for two randomly selected clients $i$ in Fig. 2. More results are shown in the Appendix. It can be seen that local smoothness $\frac{\|\nabla f_i(x_t) - \nabla f_i(x_{t-1})\|}{\|x_t - x_{t-1}\|}$ positively correlates with gradient norm $\|\nabla f_i(x_t)\|$ instead of a constant, which satisfies weaker Assumption 2. Besides, the gradient noise meets heavy-tailed distribution, i.e., Assumption 3, rather than light-tailed Gaussian distribution.

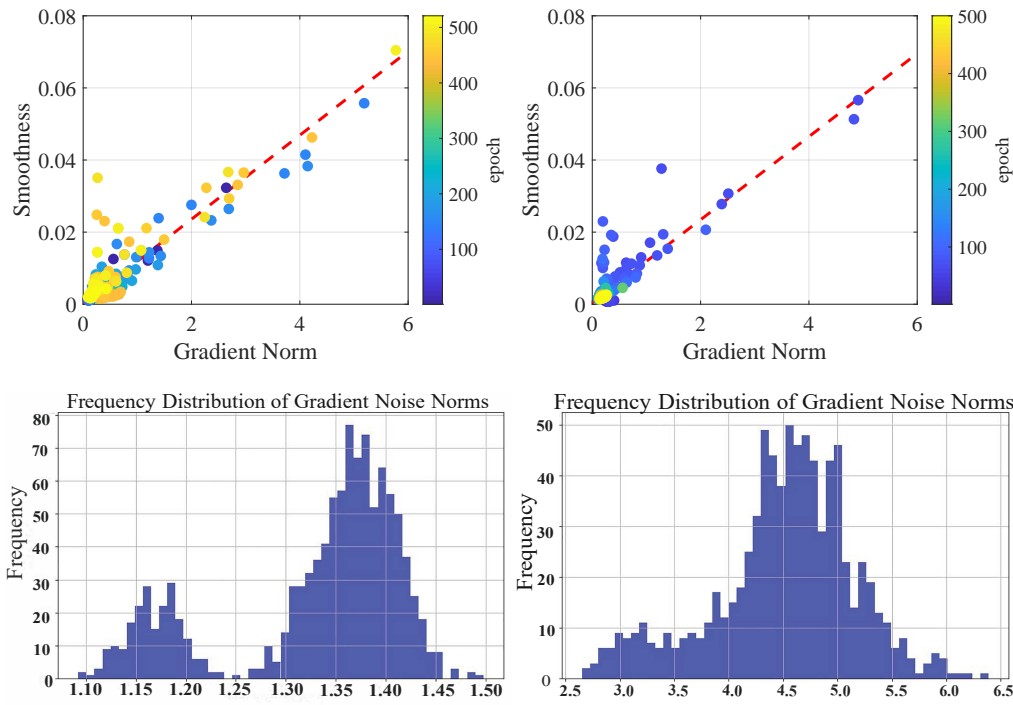

Figure 2: Gradient norm vs estimated Lipschitz smoothness (left) and distributions of the gradient noises (right) during training a stacked LSTM [26] on the Shakespeare dataset.

**Hyperparameter selection.** We conducted ablation experiments on hyperparameters $\gamma$, $\lambda$, $b$ and $K$ as shown in Fig. 3 and Fig. 6 in the Appendix. When global learning rate $\gamma = 0.2$ or $0.3$ and $\lambda = 3.0$, our `FedCBG` algorithm performs better than other choices. The performance of $b = 100$ and $K = 10 \times n_i/b$ exceeds that of other values.

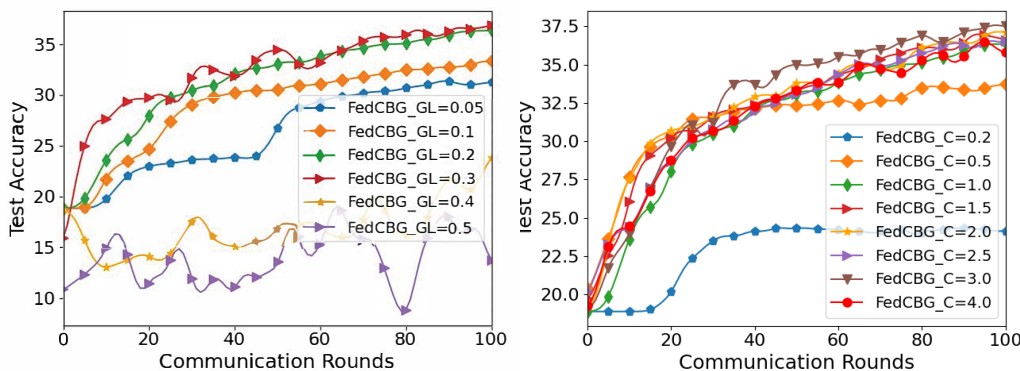

Figure 3: Test accuracy with different global learning (GL) rate (left) and clipping (C) parameter (right) on the Shakespeare dataset.

Table 3: Comparison of the training loss (TLoss.), testing classification accuracy (TAcc.) and the number of communication rounds (Round) to reach target test accuracy (84.5% for CIFAR-10, 45.0% for CIFAR-100 and 35.5% for Shakespeare datasets) in FL with heavy-tailed noise on various datasets.

| Datasets | Evaluation | CIFAR-10 | CIFAR-100 | Shakespeare |
|---|---|---|---|---|
| | TLoss | 0.16 | 3.25 | 3.17 |
| PR | TAcc. (%) | 83.1 | 42.2 | 34.8 |
| | Round | 282 (3.1×) | 412 (1.9×) | 219 (2.2×) |
| | TLoss | 0.10 | 3.16 | 3.18 |
| PI | TAcc. (%) | 84.0 | 42.8 | 35.2 |
| | Round | 189 (2.1×) | 346 (1.6×) | 178 (1.8×) |
| | TLoss | 0.12 | 3.20 | 3.52 |
| FedAvg | TAcc. (%) | 83.8 | 42.0 | 32.0 |
| | Round | 201 (2.3×) | 409 (1.9×) | 268 (2.7×) |
| | TLoss | **0.07** | **3.00** | **3.04** |
| FedCBG | TAcc. (%) | **85.6** | **44.2** | **36.5** |
| | Round | **89** | **221** | **98** |

**Experimental details.** Firstly, we choose $\eta = 1$, $\lambda = 3.0$, $\gamma = 0.3$, $K = n_i/b$ and $b = 100$ to train all the methods. Device distributions are non-IID. We use 100 randomly selected clients to train the model and the remaining 39 clients to test the model performance, which can quantify the performance gap on unseen client distributions. We report the average experimental results of 10 random initializations in Table 3. Secondly, we also conducted an experimental comparison based on the well-chosen hyperparameters and more results are shown in the Appendix. Table 3 shows that FedCBG can achieve 1.6-3.1 times gains over the competitors on all the tasks including vision and text models, which verifies the validity of our analysis: our FedCBG converges faster and performs better generalization ability than baselines on unseen client distributions.

# 7 Conclusions and Future Work

In this paper, we study clipped SGD for heavy-tailed noise. We prove a tighter optimization upper bound and the advanced generalization bound in high probability under weaker conditions. We extend our analysis to the federated setting based on our batch gradient variance bound and propose a FedCBG algorithm, which first achieves the high-probability convergence rate matching the lower bound and the first high-probability generalization bound. In future work, we will explore the role of recursive momentum [8] and minimax optimization [27, 28, 2] in heavy-tailed scenarios.

# 8 Acknowledgements

We want to thank the anonymous reviewers for their valuable suggestions and comments. This work was supported by the National Key Research and Development Program of China (No. 2023YFF0906204), National Natural Science Foundation of China (No. 62276182), Peng Cheng Lab Program (No. PCL2023A08), and the Ministry of Education, Singapore, under its Academic Research Fund Tier 1 (RG101/24).

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
