# OpenReview forum: "Tight High-Probability Bounds for Nonconvex Heavy-Tailed Scenario under Weaker Assumptions"
_NeurIPS.cc/2025/Conference — NeurIPS 2025 poster_

### Official Review · Reviewer_6QjF · 2025-06-30

**Clarity:** 3
**Significance:** 3
**Originality:** 3
**Rating:** 4
**Confidence:** 3

**Summary:**

This paper provides the convergence bounds and the generalization error bounds for clipped-SGD, considering the $(L_0,L_1)$-smoothness and the heavy-tailed noise assumption. The authors also extend the clipped-SGD to federated learning and provide both the convergence bounds and the experimental results to illustrate the effectiveness.

**Questions:**

Is it possible to obtain a convergence bound with $\log(T/\delta)$ order, as the one in [1]? Or is it possible to drop the logarithm factor and match the lower bound?

[1] Ashok Cutkosky and Harsh Mehta. High-probability bounds for non-convex stochastic optimization with heavy tails. Advances in Neural Information Processing Systems, 34:4883–4895, 2021.

**Ethical Concerns:**

["NO or VERY MINOR ethics concerns only"]

**Final Justification:**

This paper provides novel convergence results for clipped SGD under $(L_0,L_1)$-smoothness and the heavy-tailed noise. I think it's a borderline paper. The strength is that the result is new. The weakness comes from that the main proof techniques still rely on the existing proof techniques of clipped SGD under heavy-tailed noise (usually on $L$-smoothness). Also, there are some minor issues regarding the presentation, including a lack of detailed proof sketches and a detailed comparison to existing works.

Overall, I think it's a borderline paper.

**Quality:**

2

**Strengths And Weaknesses:**

**Strengths**:

a. The theoretical results are new and interesting, which are beneficial for the optimization community. There are no convergence bounds and generalization error bounds for the $(L_0,L_1)$-smoothness and the heavy-tailed noise assumption. Also, the improvement of logarithm order in the convergence bound is interesting.

b. The proof looks sound. Although the proof borrows some techniques from the convergence analysis of heavy-tailed noise and $(L_0,L_1)$-smoothness, it's OK. Also, there are some new proof techniques.

c. It's also interesting to study the federated learning scenario, which is rarely considered in the classical optimization field.

**Weaknesses**:

a. Although the results are new, the contribution is a bit incremental. It's well known that the convergence results for clipped-SGD have been derived under standard $L$-smoothness and the heavy-tailed noise assumption. When relaxing to $(L_0,L_1)$-smoothness, the additional gradient inside the smooth parameter can be cancelled by a proper selection of the step-size. One of the central proof techniques still relies on the classical results for the heavy-tailed noise, such as the result in [29] to bound the two types of noise variance, as indicated in Eq. (15) on Page 16.

b. If there are some new estimation techniques due to the $(L_0,L_1)$-smoothness, it would be better to elaborate more in the proof sketch. The proof sketch lacks details.

c. The results for FL seem not very novel in comparison with the ones for CL. I will regard the theoretical results for CL as the major contribution in this paper, while the ones for FL as extension results. If there are some essential differences in the results or the proof techniques, it may be better to present them clearly. Otherwise, the part for FL can be compressed or moved to the appendix while adding more proof details for the results of CL in the main body of the paper.

d. Although there is an improvement on the logarithm order in the convergence bound, the improvement can be considered a bit weak since the dominated order is the polynomial dependency on $T$ instead of the logarithm order.

---

> ### Author Rebuttal · Authors · 2025-07-31
>
> ### **Our analysis is non-trivial.**
>
> $\bullet$ **Based on the bounded gradient assumption and the criterion in expectation, it is trivial to extend to $(L_0, L_1)$-smoothness by choosing the step size.** There are works [4, 5] to extend the standard smoothness to the $(L_0, L_1)$-smoothness in the heavy-tailed noise. They are all based on criterion in expectation and even rely on bounded stochastic gradients. The criterion in expectation helps to tighten the upper bound of the error. For example, there is $\mathbb{E}_t\|\theta_t^a\|^2 \leq 10\sigma^p\lambda^{2-p}$ but only $\|\theta_t^a\|^2 \leq 4\lambda^2$ without expectation operation, where $\lambda = \mathcal{O}(T^{\frac{1}{3p-2}})$. The bounded stochastic gradient can be used to handle unbounded variance by decomposing it into $p$-th moment and gradient norm. However, **based on the high-probability analysis, it is non-trivial to extend the standard smoothness to the $(L_0, L_1)$-smoothness.**  On the one hand, some error terms in high-probability criterion are larger than those in expectation criterion. For example, there is $\mathbb{E}_t\|\theta_t^a\|^2 \leq 10\sigma^p\lambda^{2-p}$ in expectation but only $\|\theta_t^a\|^2 \leq 4\lambda^2$ in high-probability. On the other hand, we remove the bounded gradient assumption. Thus, it is difficult to handle $(L_0, L_1)$-smoothness only by choosing step sizes. Nevertheless, we provide an improved high-probability analysis only under $(L_0, L_1)$-smoothness and heavy-tailed noise assumptions.
>
> $\bullet$ Eq. (15) is a basic tool for handling heavy-tail noise. Many works [1, 2, 3] also depend on such results. Although our analysis depends on Eq. (15), there are many differences compared with [29].
>
> **New challenges**
>
> $(L_0, L_1)$-smoothness leads to the following challenges: Firstly, obtaining an upper bound on the gradient $\|\nabla F_S(x_t)\|$ is difficult because it is hidden in an inequality. Secondly, we need to construct a new martingale difference sequence to handle the term $L_1\|\nabla F_S(x)\|$ in $(L_0, L_1)$-smoothness. The two factors complicate choosing hyper-parameters $\lambda$ and $\eta$.
>
> **Novel contributions**
>
> By induction, we deduce that the gradient $\|\nabla F_S(x_t)\|$ needs to satisfy the quadratic inequality,  $\|\nabla F_S(x_t)\| \leq \sqrt{4(L_0 + L_1\|\nabla F_S(x_t)\|)\Delta_1}$, so that the gradient $\|\nabla F_S(x_t)\|$ can be controlled under the $(L_0, L_1)$-smoothness when $\lambda$ is greater than a constant. In addition, we construct a new martingale difference sequence, $\sum_{t=0}^{l-1}(2\eta - L_0\eta^2 - L_1\eta^2\|\nabla F_S(x_t)\|)\langle X_t, \theta_t^a\rangle$, where $X_t = \left\\{\begin{matrix}-\nabla F_S(x) & \text{if } \Delta_t \leq 2\Delta_1 \newline 0 &\text{otherwise} \end{matrix}\right.$, and bound it by the Freedman’s inequality. Finally, by solving the inequality w.r.t. the clipping parameter $\lambda$, we obtain a better parameter setting for step size $\eta$, thereby achieving a better convergence rate $\mathcal{O}(T^{\frac{2-2p}{3p-2}}\log^\frac{2p-2}{2p-1}\frac{T}{\delta})$, compared with$\mathcal{O}(T^{\frac{2-2p}{3p-2}}\log^2\frac{T}{\delta})$ in work [29].
>
> **Detailed comparisons between our paper and [29].**
>
> (1) Different technologies. Under standard smoothness, [29] uses induction and construct a martingale difference sequence
>
> $$U_s^t = \left\\{\begin{matrix} 0 & s = 0\newline \text{sgn} \left(\sum_{i=1}^{s-1} U_i^t\right) \frac{\langle \sum_{i=1}^{s-1}\beta^{t-i}\theta_i^a, \beta^{t-s}\theta_s^a\rangle}{\|\sum_{i=1}^{s-1}\beta^{t-i}\theta_i^a\|} & s\neq 0 \text{ and } \sum_{i=1}^{s-1}\beta^{t-i}\theta_i^a \neq 0\newline 0 & s\neq 0 \text{ and }  \sum_{i=1}^{s-1}\beta^{t-i}\theta_i^a = 0\end{matrix}\right.$$
>
> to bound errors, where $s \in \{0\} \cup [t]$ and $\beta = 1 - T^{-\frac{p}{3p-2}}$, while we construct a new martingale difference sequence $\sum_{t=0}^{l-1}(2\eta - L_0\eta^2 - L_1\eta^2\|\nabla F_S(x_t)\|)\langle X_t, \theta_t^a\rangle$ and solve a quadratic inequality about gradient $\|\nabla F_S(x_t )\|$ to bound errors.
>
> (2) Different intermediate results. As we propose new technologies above to deal with $(L_0, L_1)$-smoothness, our inductive Lemma 6 is very different from Lemma 8 in [29], which plays a key role in proving convergence rate. Due to different martingale difference sequences, each term in our Lemma 6 needs to be rebounded rather than simply extending the results in Lemma 8 in [29].
>
> (3) Better convergence rates under weaker assumptions. [29] achieves the convergence rate $\mathcal{O}(T^{\frac{2-2p}{3p-2}}\log^2\frac{T}{\delta})$ but depends on standard smoothness of $F_S(x)$. Our analysis is based on the general $(L_0, L_1)$-smoothness and achieves better convergence rate $\mathcal{O}(T^{\frac{2-2p}{3p-2}}\log^{\frac{2p-2}{2p-1}}\frac{T}{\delta})$ as shown in Table 1.
>
> (4) Generalization. We prove the generalization bounds for CL and FL but the work [29] did not discuss this issue. To the best of our knowledge, our paper is the first work that provides the generalization bound under the heavy-tailed noise and $(L_0, L_1)$-smooth setting.
>
> (5) Federated case. We propose the novel FedCBG algorithm for heavy-tailed federated setting. More importantly, we propose novel error estimations as shown in Lemmas 1 and 2 to obtain the nearly optimal convergence rate. Thus, it is non-trivial to extend our technology to the federated setting. But [29] did not consider the federated case.
>
> [1] Nguyen. Improved convergence in high probability of clipped gradient methods with heavy tailed noise. NeurIPS.
>
> [2] Li. High probability analysis for non-convex stochastic optimization with clipping. ECAI 2023.
>
> [3] Cutkosky. High-probability bounds for non-convex stochastic optimization with heavy tails. NeurIPS, 2021.
>
> [4] Liu. Nonconvex Stochastic Optimization under Heavy-Tailed Noises: Optimal Convergence without Gradient Clipping. ICLR.
>
> [5] Zhang. Why are adaptive methods good for attention models? NeurIPS, 2020.
>
> ### **Estimation techniques for $(L_0, L_1)$-smoothness.**
>
> **Challenges**
>
> $(L_0, L_1)$-smoothness leads to the following challenges:
>
> 1. Obtaining an upper bound on the gradient $\|\nabla F_S(x_t)\|$ is difficult because it is hidden in an inequality.
> 2. We need to construct a new martingale difference sequence to handle the term $L_1\|\nabla F_S(x)\|$ in $(L_0, L_1)$-smoothness.
>
> These two factors complicate the selection of hyper-parameters $\lambda$ and $\eta$.
>
> **Novel estimation techniques**
>
> 1. By induction, we deduce that the gradient $\|\nabla F_S(x_t)\|$ needs to satisfy the quadratic inequality $\|\nabla F_S(x_t)\| \leq \sqrt{4(L_0 + L_1\|\nabla F_S(x_t)\|)\Delta_1}$ so that the gradient $\|\nabla F_S(x_t)\|$ can be controlled when $\lambda$ is greater than a constant.
> 2. We construct a new martingale difference sequence $\sum_{t=0}^{l-1}(L_1\eta^2\|\nabla F_S(x_t)\| - \eta)\langle \nabla F_S(x_t), \theta_t^a\rangle$ and bound it by the Freedman’s inequality.
> 3. By solving the inequality w.r.t. $\lambda$, we obtain a better parameter setting about $\eta$, thereby achieving a better convergence rate $\mathcal{O}(T^{\frac{2-2p}{3p-2}}\log^\frac{2p-2}{2p-1}\frac{T}{\delta})$.
>
> We will add these details to the revised version.
>
> **The first generalization result under $(L_0, L_1)$-smoothness**
>
> To the best of our knowledge, there is no work to analyze the generalization bound under $(L_0, L_1)$-smoothness. We propose a new analytical method base on uniform convergence to quantify generalization ability only under weaker assumptions (i.e., $(L_0, L_1)$-smoothness and heavy-tailed noise), which is shown in Section 4.2.
>
> ### **It is non-trivial to extend the analysis in CL to FL.**
>
> Extending CL to FL will cause the error term $A \leq \mathcal{O}((mKT)^\alpha), \alpha > 0$ in Eq. (66), which restricts $\gamma \leq \mathcal{O}((mKT)^{\frac{-p-1}{3p-2}})$. A nearly optimal convergence rate requires $\gamma \geq \mathcal{O}((mKT)^{\frac{-p}{3p-2}})$, resulting in a contradiction. **Firstly**, we proposed the clipped mini-batch gradient and analyzed the batch gradient variance under heavy-tailed setting in Lemma 1, which open the door to analyze the mini-batch methods in the heavy-tailed noise. **Secondly**, different from Eq. (15) in the CL, we prove the upper bound for errors $\epsilon_t^a$ and $\epsilon_t^b$ as shown in Lemma 2. They are the new results under the federation setting. **Thirdly**, we propose a "sum-aggregation" as shown in Algorithm 2 to facilitate our proof. **Finally**, $(L_0, L_1)$-smoothness makes the gradient implicit in Ineq. (61) and we handle it by solving a quadratic inequality. Thus, such expansion is non-trivial. To address your concerns, we will add a discussion in our revised version.
>
> ### **About the logarithm factor $\log(\frac{T}{\delta})$.**
>
> It is possible to obtain a convergence bound with $\log(T/\delta)$ order as in [1]. In fact, the work [29] achieves this result. Note that they are all based on the criteria $\frac{1}{T}\sum_{t=1}^T\|\nabla F_S(x_t)\|$. According to Jensen's inequality, our convergence rate is $\mathcal{O}(T^\frac{1-p}{3p-2}\log^\frac{p-1}{2p-1}\frac{T}{\delta})$ based on this criteria, which is better than [6] [29] w.r.t. $\log(T/\delta)$ order due to $\frac{p-1}{2p-1} \in (0, \frac{1}{3}]$. Although this improvement is not significant, it is achieved under the weaker $(L_0, L_1)$-smoothness assumption.
>
> In the heavy-tailed noise, the lower bound in expectation is $\mathcal{O}(T^{\frac{2-2p}{3p-2}})$ [2] and it proves that global clip (GClip) algorithm can achieves this lower bound. But in high-probability analysis, the lower bound is unknown, and it is not clear whether the logarithm factor can be completely removed. This is also one of our future research directions.
>
> [6] Ashok. High-probability bounds for non-convex stochastic optimization with heavy tails. NeurIPS, 2021.
>
> [7] Zhang. Why are adaptive methods good for attention models?. NeurIPS, 2020.

---

> > ### Comment · Reviewer_6QjF · 2025-08-04
> >
> > Thank you for your reply. I appreciate the detailed response and agree that extending from $L$-smoothness to $(L_0,L_1)$-smoothness can be non-trivial. My main concern is whether such an extension constitutes a substantial contribution, particularly since the extension of SGD to $(L_0,L_1)$-smoothness has been shown to be essentially similar in [1]. However, the result in this paper is novel.
> >
> > I will make my final decision after further consideration.
> >
> > [1] Léon Bottou, Frank E. Curtis, and Jorge Nocedal. “Optimization Methods for Large-Scale Machine Learning”. In: SIAM Review 60.2 (2018), pp. 223–311.

---

> > > ### Author Response · Authors · 2025-08-06
> > >
> > > Thanks for your comments. Here is our further clarification.
> > >
> > > **Extension to $(L_0, L_1)$-smoothness** ***under weaker assumptions*** **is challenging and fundamentally different from the proofs in [1].**
> > >
> > > Firstly, our analysis faced not only $(L_0,L_1)$-smoothness challenge but also other *weaker assumption* challenges. Although pioneering work [2] has shown that the extension of SGD to $(L_0, L_1)$-smoothness is similar to the proof in [1], the work [2] depends on a strong assumption $\\|\nabla F_S(x) - \nabla f(x;\xi)\\| \leq \sigma$ and bounded gradient assumption. On the contrary, our analysis only needs the heavy-tailed noise assumption $\mathbb{E}\\|\nabla F_S(x) - \nabla f(x;\xi)\\|^p \leq \sigma^p$, which is weaker than $\\|\nabla F_S(x) - \nabla f(x;\xi)\\| \leq \sigma$ and bounded gradient assumption. Thus, our analysis is more difficult than [1, 2] rather than being similar to existing analysis.
> > >
> > > Secondly, our proof is fundamentally different from the proofs in [1, 2]. The proofs in the works [1, 2] are based on the expectation criterion, while ours is based on the high-probability criterion. Specifically, we use induction, construct new martingale difference sequences and solve a quadratic inequality in our analysis, which eliminates $\\|\nabla F_S(x) - \nabla f(x;\xi)\\| \leq \sigma$ and the bounded gradient assumption and relies only on the heavy-tail noise assumption. These new techniques are not used in the proofs of [1, 2].
> > >
> > > In addition to extending to $(L_0, L_1)$-smoothness, we propose a new FedCBG algorithm to match the optimization lower bound in FL without the bounded gradient assumption, which is another challenge and substantial contribution of our paper.
> > >
> > > [1] Léon Bottou, Frank E. Curtis, and Jorge Nocedal. “Optimization Methods for Large-Scale Machine Learning”. In: SIAM Review 60.2 (2018), pp. 223–311.
> > >
> > > [2] Zhang J, He T, Sra S, et al. Why Gradient Clipping Accelerates Training: A Theoretical Justification for Adaptivity[C]. International Conference on Learning Representations.

---

> > > ### Author Response · Authors · 2025-08-09
> > >
> > > We sincerely appreciate your review and valuable questions that improved our work. We hope we have adequately addressed all of the reviewer’s concerns.
> > >
> > > As mentioned in our response, extending to $(L_0, L_1)$-smoothness under weaker assumptions presents unique challenges that differ fundamentally from existing proofs. Besides, we propose a new FedCBG algorithm for the federated setting, which achieves an advanced high-probability bound under weaker assumptions. Given this context, would it be possible for the reviewer to adjust the score accordingly? If you have any further questions, we would be delighted to discuss them with you.
> > >
> > > Thank you for your time and consideration.

---

### Official Review · Reviewer_AasK · 2025-06-30

**Clarity:** 2
**Significance:** 3
**Originality:** 2
**Rating:** 5
**Confidence:** 4

**Summary:**

This paper analyzes the Clipped SGD method under more relaxed assumptions of $(L_0,L_1)$-smoothness and heavy-tailed noise for nonconvex functions. Current literature tackles only one of the cases but two simultaneously. The authors extend the results to a Federated setting, providing the FedCBG method with similar convergence guarantees. On top of the convergence results, the authors provide generalization bounds for the Clipped SGD and FedCBG methods. Importantly, the convergence results are optimal in the number of iterations $T$.

**Questions:**

- I suggest moving assumptions 2 and 3 to the introduction, as they are used a lot before formally providing them. This would improve the clarity of the presentation.

- There is a concurrent work [Chezhegov, Savelii, et al. "Convergence of Clipped-SGD for Convex $(L_0, L_1) $-Smooth Optimization with Heavy-Tailed Noise." arXiv preprint arXiv:2505.20817 (2025)]. They consider the same setting assumptions but in the convex regime. It would be worth mentioning the paper and comparing the results, though it is not necessary. Interestingly, they obtain the optimal rate only after $\mathcal{O}(1/\delta)$ iterations. There is no such restriction in this work. Can the authors provide an intuition why it is so if possible?

- There is no gradient-similarity-type assumption in the theory of the FedCBG method, while vanilla FedAvg's rate does depend on such a quantity. How is it avoided here? Is it hidden somewhere?

- Is the power of $\log$ terms in the rates optimal, or can they be improved further?

- What is the difference between tables 1 and 4?

- How is the power of $\log$ terms improved? Does it only come from a more careful choice of the hyperparameters, like stepsize?

- How are problem parameters like $L_0, L_1, \Delta_1$ present in the rates? Can the authors provide explicit rates? Right now, I suspect that the rates are worse than what I expect in the nonconvex regime (see, e.g., [Hübler et al. "Parameter-agnostic optimization under relaxed smoothness." AISTATS, 2024].

- What is the value of $p$ in the experiments? Can the authors estimate it to ensure that the noise is indeed heavy-tailed with $p\in(1,2]$?

- Table 3 contains the results on CIFAR10, which is not heavy-tailed as far as I know. Therefore, the caption of Table 3 should be clarified.

- What algorithm is used to obtain Figure 2?

- What does "recursive momentum" mean?

- What is the difference between the settings of the two left and two right plots in Figure 2?

**Ethical Concerns:**

["NO or VERY MINOR ethics concerns only"]

**Final Justification:**

The authors provided clarifications and addressed all my concerns by providing very detailed answers. I believe that this work is a significant contribution to the field of optimization, in particular, the convergence of gradient-based algorithms with heavy-tailed noise. The authors tackle a challenging case of non-convex $(L_0,L_1)$-smooth functions. The obtained results are optimal w.r.t. several quantities in the rate, demonstrating the tightness of the analysis. Therefore, I increase my score to 5, and support acceptance of this work.

**Limitations:**

The authors provided some discussion in section 7, though it is very limited. In particular, it is not clear what "recursive momentum" means and which part of the complexity results they aim to reduce. I suggest also including

- Generalization to symmetric $(L_0, L_1)$-smoothness.

- Dependency on the problem parameters $L_0,L_1$.

in the limitations.

**Quality:**

3

**Strengths And Weaknesses:**

***Strengths:***

- Strong theoretical guarantees for the Clipped SGD algorithm with an optimal dependency on the number of iterations $T$, only with a logarithmic factor depending on the failure probability $\delta$.

- I went through the proofs, and they look correct to me. The idea is to inductively bound the martingale differences.

- The analysis does not depend on the gradient bound, which is an improvement over some earlier works.

- Extension to the federated setting and an algorithm FedCBG, which also converges under relaxed smoothness and heavy-tailed noise with a competitive performance in practice.

***Weaknesses:***

- The results seem to be suboptimal in $L_0$ and $L_1$. It looks like we would get the same rates when using SGD with a fixed stepsize. In other words, there is no advantage to using the Clipped SGD.

- The related works section can be improved further. Several important works that also study heavy-tailed noise assumption are not discussed, e.g., [Sadiev et al. "High-probability bounds for stochastic optimization and variational inequalities: the case of unbounded variance." ICML, 2023.], [Gorbunov et al. "High-probability convergence for composite and distributed stochastic minimization and variational inequalities with heavy-tailed noise." ICML 2024], one of them studies the federated setting as well.

- The current theoretical results in the main paper are hard to parse. The dependency on other problem parameters, e.g., $L_0, L_1$ is missing. Several quantities that are used in the theorems, e.g., $\Delta_1$ are not defined, or it is not clear how they affect the rate, e.g., $R$ or $\mathcal{R}$. Similar claims are applied to the results presented in the table, to the proof sketches (what are the constants $A_{1,2,3}$ ?), and to the rate of FedCBG.

- The convergence is given for a local version of $(L_0, L_1)$-smoothness, i.e. the smoothness holds when $\|x-y\|\le 1/L_1$ (assymetric $(L_0,L_1)$-smoothness). It would be interesting to see the convergence results when a more general class of symmetric $(L_0,L_1)$-smooth functions is considered, i.e., for all $x,y\in\mathbb{R}^d$ we have $\\|\nabla f(x)-\nabla f(y)\\| \le (L_0+L\_1\max\_{u\in[x,y]}\\|\nabla f(u)\\|)\cdot \\|x-y\\|$.

---

> ### Author Rebuttal · Authors · 2025-07-31
>
> #### **About weaknesses**
>
> $\bullet$ Under the bounded variance assumption, SGD can achieve convergence rate $\mathcal{O}(T^{-1/4})$, which is consistent with our results when $p = 2$. However, under the heavy-tailed noise assumption, clipped SGD performs better than SGD. Clipped SGD converges under heavy-tailed noise, while standard SGD may diverge [1].
>
> $\bullet$ Thanks for your comments. The work [2] proposed a high-probability analysis for clipped SGD and achieved the convergence rate of $\mathcal{O}(T^{\frac{1-\alpha}{\alpha}})$. Although it eliminates the bounded gradient assumption, its convergence rate is worse than advanced one in [3]. The work [4] considers the distributed setting but under the convexity assumption, which limits its scope of application. To address your concerns, we will discuss the two works in our revised version.
>
> $\bullet$ Thanks for your comments. $\Delta_t = F_S(x_t) - F^*$ and $R$ and $\mathcal{R}$ are constants. They do not affect the convergence rate results.  $A_1$ $A_2$ $A_3$ are defined in Eq. (88). For readability and space limitations, we put some detailed parameter dependencies in the Appendix. For example, $\lambda = \max\{\big(\frac{64}{\mathcal{R}\rho}\big)^{\frac{1}{2p-1}}\sigma^{\frac{2p}{2p-1}}T^{\frac{1}{3p-2}}, \frac{(2\Delta_1)^{\frac{1}{p}}\sigma T^{\frac{1}{3p-2}}}{(\mathcal{R}\rho^2)^{\frac{1}{p}}}, 40^{\frac{1}{p}}{T^{\frac{1}{3p-2}}}\rho^{-\frac{2}{p}}\sigma (1+\frac{L_1\Delta_1}{32\mathcal{R}})^\frac{2}{p}, 4R\}$ as shown in Eq. (31) in Line 603, where $\mathcal{R} = L_0 + 2(L_1 + 1)R$, $R = \frac{1}{2}(L_0 + (4L_1+1)\Delta_1)$ and $\rho = \max\{\log\frac{4T}{\delta}, 1\}$. To address your concerns, we will add them to our main paper.
>
> $\bullet$ Thanks for your comments. We will consider to study the convergence properties and generalization bounds under symmetric $(L_0, L_1)$-smoothness assumption in future work.
>
> [1] Marshall N, Xiao K L, Agarwala A, et al. To Clip or not to Clip: the Dynamics of SGD with Gradient Clipping in High-Dimensions[C]//The Thirteenth International Conference on Learning Representations.
>
> [2] Sadiev A, Danilova M, Gorbunov E, et al. High-probability bounds for stochastic optimization and variational inequalities: the case of unbounded variance[C]//International Conference on Machine Learning. PMLR, 2023: 29563-29648.
>
> [3] Liu Z, Zhang J, Zhou Z. Breaking the lower bound with (little) structure: Acceleration in non-convex stochastic optimization with heavy-tailed noise[C]//The Thirty Sixth Annual Conference on Learning Theory. PMLR, 2023: 2266-2290.
>
> [4] Gorbunov E, Sadiev A, Danilova M, et al. High-probability convergence for composite and distributed stochastic minimization and variational inequalities with heavy-tailed noise[J]. arXiv preprint arXiv:2310.01860, 2023.
>
> #### **About Questions**
>
> $\bullet$ Thanks for your comments. We will reposition Assumptions 2 and 3 in our revised version.
>
> $\bullet$ Extending the results in the convex case to the non-convex case remains a significant challenge. Although the convex case can achieve the optimality, the non-convex problem is very difficult due to the existence of multiple local optimal points. The convergence rate we have proved is state-of-the-art so far and whether it can be achieved optimally is still an open question.
>
> $\bullet$ This is due to the novelty of our proof technique. We decompose the quadratic term into errors $\epsilon_t^a$ and $\epsilon_t^b$ in the inductive Lemma 3. Then, we construct new martingale difference sequences and use the Freedman’s inequality, which removes the gradient-similarity-type assumption.
>
> $\bullet$ There is no lower bound for high-probability analysis, it is unknown whether the power of $\log$ terms can be improved.
>
> $\bullet$ Table 4 is more detailed. Table 4 contains more related work and compares several results based on the criteria in expectation. Due to space limitations, we put the main compared methods in the main paper.
>
> $\bullet$ The improvement come from our new martingale difference sequences and a more careful choice of hyper-parameters. We construct a new martingale difference sequence, $\sum_{t=0}^{l-1}(2\eta - L_0\eta^2 - L_1\eta^2\|\nabla F_S(x_t)\|)\langle X_t, \theta_t^a\rangle$, where $X_t = \left\\{\begin{matrix}-\nabla F_S(x) & \text{if } \Delta_t \leq 2\Delta_1 \newline 0 &\text{otherwise} \end{matrix}\right.$. The new sequence requires us to solve new inequalities about the clipping parameter $\lambda$. Then, we can obtain a better parameter setting about step size $\eta$, thereby achieving a better convergence rate $\mathcal{O}(T^{\frac{2-2p}{3p-2}}\log^\frac{2p-2}{2p-1}\frac{T}{\delta})$.
>
> $\bullet$ Under the bounded variance assumption, the work [5] can achieve convergence rate $\mathcal{O}(T^{-1/4})$, which is consistent with our results when $p = 2$. The work [5] needs the bounded variance assumption but our results are based on heavy-tailed noise assumption. The explicit rate for CL is $\mathcal{O}(\max\\{\frac{\sigma^{\frac{2p}{2p-1}}{\mathcal{R}}^{p}}{T^{\frac{2-2p}{3p-2}}}, \frac{\sigma(2\Delta_1)^{\frac{1}{p}}}{\mathcal{R}^{\frac{1}{p}}T^{\frac{2p-2}{3p-2}}}, \frac{\sigma(1+\frac{L_1\Delta_1}{\mathcal{R}})^\frac{2}{p}}{T^\frac{2p-2}{3p-2}}\\}\log^{\frac{2p-2}{2p-1}}\frac{T}{\delta})$. Similarly, we can also get the results under FL. To address your concerns, we will move them in our revised version.
>
> $\bullet$ The heavy-tailed phenomenon can be verified by the two right plots on the right side of Figure 2. To address your concerns, we will estimate the value of $p$ in our revised version.
>
> $\bullet$ In the federated learning, CIFAR-10 occasionally performs heavy-tailed phenomenon. Similar experiments can be found in [6]. To address your concerns, we will delete "heavy-tailed noise" in the caption of Table 3.
>
> $\bullet$ We use the FedAvg algorithm to obtain Figure 2.
>
> $\bullet$ "recursive momentum" was proposed by the work [7], which is a technique that can achieve variance reduction using only a single stochastic sample.
>
> $\bullet$ There is no difference between the settings of the two left and two right plots in Figure 2.
>
> [5] Hübler et al. "Parameter-agnostic optimization under relaxed smoothness." AISTATS, 2024
>
> [6] Yang H, Qiu P, Liu J. Taming fat-tailed (“heavier-tailed” with potentially infinite variance) noise in federated learning[J]. Advances in Neural Information Processing Systems, 2022, 35: 17017-17029.
>
> [7] Cutkosky A, Orabona F. Momentum-based variance reduction in non-convex sgd[J]. Advances in neural information processing systems, 2019, 32.

---

> > ### Comment · Reviewer_AasK · 2025-08-03
> > **Response to the rebuttals**
> >
> > I thank the reviewers for detailed replies. I still have some minor comments.
> >
> > - What I meant by comparing with the rate of SGD is that in the light-tailed case SGD has some dependency on the problem constants like $L_0$ and $L_1$. Can the authors compare their results in the heavy-tailed case with the results of SGD/clipped-SGD in the light-tailed case from this perspective? This is important to understand if there is still room for improvement.
> >
> > - I agree that the works I mentioned have other limitations, but the authors should still revise their literature review to make the work stronger. Therefore, I encourage the authors to provide a discussion on how they aim to incorporate those papers.
> >
> > - I suggest not hiding some works in the main body and moving Table 4 to the main part to improve readability of the work.
> >
> > - Can the authors provide estimates of $p$ during the discussion? Can the authors provide a discussion on how they are going to estimate it?

---

> > > ### Author Response · Authors · 2025-08-06
> > > **Responses to Reviewer AasK [1/2]**
> > >
> > > Thanks for your comments. Here is our further clarification.
> > >
> > > ### $\bullet$**Comparisons with SGD/clipped SGD about $L_0$ and $L_1$.**
> > >
> > > For a fair comparison, we consistently use the gradient norm as the evaluation criterion in this response.
> > >
> > > **The dominated order, polynomial dependency on $T$ in our analysis, is optimal.** According to our analysis, the clipped SGD has an explicit rate is $\mathcal{O}(\max\\{\frac{\sigma^{\frac{p}{2p-1}}{\mathcal{R}}^{\frac{p}{2}}}{T^{\frac{p-1}{3p-2}}}, \frac{\sigma^\frac{1}{2}\Delta_1^{\frac{1}{2p}}}{\mathcal{R}^{\frac{1}{2p}}T^{\frac{p-1}{3p-2}}}, \frac{\sigma^\frac{1}{2}(1+\frac{L_1\Delta_1}{\mathcal{R}})^\frac{1}{p}}{T^\frac{p-1}{3p-2}}\\}\log^{\frac{p-1}{2p-1}}\frac{T}{\delta})$ under the heavy-tailed setting. When $p=2$, i.e., under bounded variance, clipped SGD can achieve the convergence rate $\mathcal{O}(\max\\{\frac{\sigma^{\frac{2}{3}}{\mathcal{R}}}{T^{\frac{1}{4}}}, \frac{\sigma^{\frac{1}{2}}\Delta_1^{\frac{1}{4}}}{\mathcal{R}^{\frac{1}{4}}T^{\frac{1}{4}}}, \frac{\sigma^\frac{1}{2}(1+\frac{L_1\Delta_1}{\mathcal{R}})^\frac{1}{2}}{T^\frac{1}{4}}\\}\log^{\frac{1}{3}}\frac{T}{\delta})$, where $\mathcal{R} = L_0 + (L_1 + 1)(L_0 + (4L_1+1)\Delta_1)$. Thus, the dominated order, polynomial dependency on $T$ in our analysis, is optimal. To address your concerns about problem constants, we will move these detailed $L_0$ and $L_1$ dependencies in the Appendix to the main text in our revised version.
> > >
> > > **Comparison with Clipped-SGD under the light-tailed noise** Clipped-SGD can achieve the convergence rate $\mathcal{O}(\frac{(\Delta_1 L_0)^{\frac{1}{4}}\sigma^{\frac{1}{2}}}{T^{\frac{1}{4}}})$ for $(L_0, L_1)$-smoothness [1, 3], which matches the lower bound $\Omega(\frac{(\Delta_1 L)^{\frac{1}{4}}\sigma^{\frac{1}{2}}}{T^{\frac{1}{4}}})$ proposed in [4]. But these results are based on *expectation criterion* rather than the *high-probability criterion* we used. Thus, this lower bound does not apply to our results.
> > >
> > > The results based on the expectation criterion are unsatisfying because the quantity measured for convergence (i.e. gradient norm or function value) is itself a random variable, and might also have a heavy tail [5]. Thus, the results based on expectation could not be quite good in each single run. On the contrary, the analysis based on high probability can provide a stronger guarantee for each individual run. To the best of our knowledge, our analysis is the first high-probability analysis under the heavy-tailed noise and $(L_0, L_1)$-smoothness assumptions.
> > >
> > > **Comparison with SGD under the light-tailed noise** SGD can achieve the convergence rate $\mathcal{O}(\max\\{\frac{(L_0 + 2L_1 G)^{\frac{1}{2}}(\Delta_1 + \sigma)^{\frac{1}{2}}}{T^{\frac{1}{2}}\delta^{\frac{1}{2}}}, \frac{G^{\frac{1}{2}}(\Delta_1 + \sigma)^{\frac{1}{2}}}{T^{\frac{1}{4}}\delta^{\frac{1}{2}}}\\})$ for $(L_0, L_1)$-smoothness, where $G := \sup\{u \geq 0|u^2 \leq 2(L_0 + 2L_1 u)\} < \infty$ [2]. The dominated order, polynomial dependency on $T$, is the same as ours, but dependency on $L_0$ and $L_1$ is different. Importantly, many works such as [6, 7, 8] show that clipped SGD performs better than SGD. For example, standard SGD may diverge under the heavy-tailed noise [8].
> > >
> > > [1] Zhang. Improved analysis of clipping algorithms for non-convex optimization.
> > >
> > > [2] Li. Convex and non-convex optimization under generalized smoothness.
> > >
> > > [3] Reisizadeh. Variance-reduced clipping for non-convex optimization.
> > >
> > > [4] Arjevani. Lower bounds for non-convex stochastic optimization.
> > >
> > > [5] Cutkosky. High-probability bounds for non-convex stochastic optimization with heavy tails.
> > >
> > > [6] Shulgin. On the Convergence of DP-SGD with Adaptive Clipping.
> > >
> > > [7] Zhang. Why Gradient Clipping Accelerates Training: A Theoretical Justification for Adaptivity.
> > >
> > > [8] Marshall. To Clip or not to Clip: the Dynamics of SGD with Gradient Clipping in High-Dimensions.

---

> > > ### Author Response · Authors · 2025-08-06
> > > **Responses to Reviewer AasK [2/2]**
> > >
> > > ### $\bullet$ **We will add the following discussion to our revised paper.**
> > >
> > > The gradient boundedness assumption is strong and needs to be removed. The existing work [9] tried to eliminate the bounded gradient assumption with a high-probability analysis for clipped SGD but only achieved the convergence rate of $\mathcal{O}(T^{\frac{1-\alpha}{\alpha}})$, which is worse than the advanced result in [10]. This fact also motivates us to propose a method or analysis that can remove the gradient boundedness assumption and maintain the advanced convergence rate.
> > >
> > > Besides the federated learning methods above, the work [11] considered the distributed heavy-tailed noise setting based on variational inequalities and achieved a tight convergence rate in the high-probability criterion. But it depends on some special structure, such as monotonicity and cocoercivity, which limits its scope of application. Thus, a tight convergence bound is urgently needed in the complex non-convex heavy-tail setting.
> > >
> > > ### $\bullet$ **We will move Table 4 to the main paper to improve readability of the work.**
> > >
> > > ### $\bullet$ **Estimation of $p$.**
> > >
> > > We will use Theorem 2.3 in the work [12] to estimate $p$. This estimation method is also used in the works [13, 14, 15] and admits an advanced convergence rate and small asymptotic variance [12, 13]. Specifically, we restate the theorem as follows.
> > >
> > > **Theorem.** $\\{X_t\\}^T_{t=1}$ be a collection of strictly $p$-stable random vectors $X_t$ and $T = T_1 \times T_2$. Define $Y_i \triangleq \sum_{j=1}^{T_1} X_{j+(i-1)T_1}$ for $i \in [1, T_2]$. Then, the estimator
> > > $$
> > > \widehat{\frac{1}{p}} \triangleq \frac{1}{\log T_1}\left(\frac{1}{T_2}\sum_{i=1}^{T_2}\log|Y_i| - \frac{1}{T}{\sum_{i=1}^T} \log|X_i|\right)
> > > $$
> > > converges to $\frac{1}{p}$ almost surely, as $T_2 \rightarrow +\infty$.
> > >
> > > Similar to the work [14], we concretize the estimation process for $p$ as follows. At iteration $t$, we first partition the set of data points ${S} \triangleq \{1, \cdots, n\}$ into many disjoint sets $S_t^j \subset {S}$ of size $b$. Thus, for $j, k = 1, \cdots, n/b$, $\cup_jS_t^j = {S}$ and $S_t^j \cap S_t^k = \emptyset$ for $j\neq k$. Then, we compute the full gradient $\nabla F_S(x_t)$ and the stochastic gradient $\nabla f(x_t;S_t^j)$ for each mini-batch. Thus, the stochastic gradient noises can be computed by $U_t^j(x_t) = \nabla F_S(x_t) - \nabla f(x_t;S_t^j)$. Vectorize each $U_t^j(x_t)$ and concatenate them to obtain a single vector $X_t$. According to the theorem above, we set $T = d n/b$ and set $T_1$ to the divisor of $T$ that is the closest to $\sqrt{T}$. Finally, we can compute the estimation $\widehat{\frac{1}{p}}$. Its reciprocal is the estimated result for $p$.
> > >
> > > [9] Sadiev. High-probability bounds for stochastic optimization and variational inequalities: the case of unbounded variance.
> > >
> > > [10] Liu. Breaking the lower bound with (little) structure: Acceleration in non-convex stochastic optimization with heavy-tailed noise.
> > >
> > > [11] Gorbunov. High-probability convergence for composite and distributed stochastic minimization and variational inequalities with heavy-tailed noise.
> > >
> > > [12] Mohammadi. On estimating the tail index and the spectral measure of multivariate α-stable distributions.
> > >
> > > [13] Gurbuzbalaban. The heavy-tail phenomenon in SGD
> > >
> > > [14] Simsekli. A tail-index analysis of stochastic gradient noise in deep neural networks
> > >
> > > [15] Zhou. Towards theoretically understanding why sgd generalizes better than adam in deep learning.

---

> > > > ### Comment · Reviewer_AasK · 2025-08-06
> > > > **Response the Authors**
> > > >
> > > > I sincerely thank the authors for such detailed answers. The rates are indeed optimal with respect to $L_0, L_1$. I encourage incorporating the missing references, discussing the optimality of the rates in relation to problem constants, and estimating $p$ in the revised manuscript. The latter is especially important to support that the heavy tail assumption is indeed a good approximation of gradient noise in practice. I am curious to see what values can be obtained by the provided estimation technique. I would highly appreciate it if the authors could post an estimate during the rebuttals.
> > > >
> > > > To summarize, I believe that this work is a significant contribution to the field of optimization, in particular, the convergence of gradient-based algorithms with heavy-tailed noise. The authors tackle a challenging case of non-convex $(L_0,L_1)$-smooth functions. The obtained results are optimal w.r.t. several quantities in the rate, demonstrating the tightness of the analysis. Therefore, I increase my score to 5.

---

> > > > > ### Author Response · Authors · 2025-08-07
> > > > >
> > > > > We thank the reviewer for the valuable feedback. We will incorporate the missing references, discuss the optimality of the rates with respect to the problem constant, and estimate $p$ in our revised version. Especially, we have performed an experiment on the Shakespeare dataset to estimate $p$, and found it to be approximately 1.7, which supports the presence of heavy-tailed noise in practice.

---

### Official Review · Reviewer_iqqA · 2025-07-02

**Clarity:** 2
**Significance:** 3
**Originality:** 2
**Rating:** 4
**Confidence:** 4

**Summary:**

This paper presents a comprehensive high-probability analysis of clipped stochastic gradient methods under weaker assumptions: (L_0, L_1)-smoothness and heavy-tailed gradient noise. The authors claim that they prove tighter high-probability convergence rates than existing work. Moreover, they design FedCBG, and establish its optimization and generalization bounds in high probability, a first in the FL literature under such weak assumptions. Experiments on CIFAR and Shakespeare datasets empirically validate both convergence and generalization improvements over baselines.

**Questions:**

1. The main theoretical results of this work rely heavily on [29], which even considers momentum. However, the distinctions between this work and [29] are not sufficiently discussed in the main text. To improve the clarity and originality of the contribution, it is recommended that the authors: (1) Provide a detailed comparison between the theoretical results of this work and those in [1], including both the differences and the respective advantages and limitations; (2) Clearly explain the specific challenges involved in extending or adapting the theoretical framework of [1] to the current setting; (3) Explicitly identify which theoretical results in this paper are novel contributions and which are direct extensions or applications of existing results from [1].

[1]	Liu Z, Zhang J, Zhou Z. Breaking the lower bound with (little) structure: Acceleration in non-convex stochastic optimization with heavy-tailed noise[C]//The Thirty Sixth Annual Conference on Learning Theory. PMLR, 2023: 2266-2290.

2. It is recommended that the authors carefully review recent advances in related work. For instance, in lines 39-40, the author claims that "only one work focuses on the heavy-tail noise problem in federated learning", but in fact, there are other works that also focus on this. For example, the work of Tao et al. [2] focuses on Byzantine-resilient federated learning in the presence of heavy-tail noise. This suggests that the related work section may not be sufficiently comprehensive and should be revised to reflect the current state of research more accurately.

[2] Tao Y, Cui S, Xu W, et al. Byzantine-resilient federated learning at edge[J]. IEEE Transactions on Computers, 2023, 72(9): 2600-2614.

3. This paper omits some important details and contains notational and typographical issues that may affect readability. For example, key symbols such as $\Delta_t$ are used in the main theorems without formal definitions. Additionally, there are problems with notational rigor, bracket mismatches, and unclear references to mathematical inequalities. It is recommended that the authors complete missing definitions, standardize the notation, and thoroughly proofread the manuscript to improve clarity, precision, and overall rigor.

**Ethical Concerns:**

["NO or VERY MINOR ethics concerns only"]

**Final Justification:**

My concerns have almost been addressed, and I raise my score accordingly.

**Limitations:**

yes

**Quality:**

3

**Strengths And Weaknesses:**

Strengths:
1. The paper makes solid theoretical advances in stochastic optimization under realistic conditions. Rigorous theoretical development with carefully designed proofs removing strong assumptions (e.g., bounded gradients), supported by martingale and inductive analyses.
2. Despite the complex technical content, the structure is well-organized, and the derivations and algorithm descriptions are clear.

Weaknesses:
1. This paper omits some key details, leading to potential ambiguity. For example, the definition of $\Delta_t$ is not formally given, but the main theoretical results, such as Theorem 1, rely on it. Therefore, it is recommended that the authors complete the key notation description that may be omitted.
2. This paper has typos or inappropriate descriptions. For example, in line 158, please confirm whether the bound is obtained based on the Jensen's inequality or should be the Cauchy-Schwarz inequality. In line 8 of Algorithm 2, $\tilde{\Delta}_t$ should have a subscript about $i$. There is a bracket error in the value of T in Theorem 4 on line 294. And so on.
3. The main theoretical results of this work rely heavily on [29] which even considers momentum optimization, but the difference between the two is not fully discussed in the main text. It is recommended that (1) the differences and advantages and disadvantages of the two theoretical results be compared in detail; (2) the difficulties in deriving the theoretical results based on [1] be explained, and specifically which results are innovative and which follow existing work.

[1]Liu Z, Zhang J, Zhou Z. Breaking the lower bound with (little) structure: Acceleration in non-convex stochastic optimization with heavy-tailed noise[C]//The Thirty Sixth Annual Conference on Learning Theory. PMLR, 2023: 2266-2290.

---

> ### Author Rebuttal · Authors · 2025-07-30
>
> ### **Detailed comparison with work [29]**
>
> #### **1. New challenges for adapting the theoretical framework of [29] to the $(L_0, L_1)$-smoothness**
>
> $(L_0, L_1)$-smoothness leads to the following challenges: Firstly, obtaining an upper bound on the gradient $\|\nabla F_S(x_t)\|$ is difficult because it is hidden in an inequality. Secondly, we need to construct a new martingale difference sequence to handle the term $L_1\|\nabla F_S(x)\|$ in $(L_0, L_1)$-smoothness. These two factors also complicate the selection of hyper-parameters such as the clipping parameter $\lambda$ and step size $\eta$.
>
> #### **2. Novel contributions and direct extensions from [29]**
>
> $\bullet$ **Novel contributions.** By induction, we deduce that the gradient $\|\nabla F_S(x_t)\|$ needs to satisfy the quadratic inequality, $\|\nabla F_S(x_t)\| \leq \sqrt{4(L_0 + L_1\|\nabla F_S(x_t)\|)\Delta_1}$, so that the gradient $\|\nabla F_S(x_t)\|$ can be controlled when $\lambda$ is greater than a constant. In addition, we construct a new martingale difference sequence, $\sum_{t=0}^{l-1}(2\eta - L_0\eta^2 - L_1\eta^2\|\nabla F_S(x_t)\|)\langle X_t, \theta_t^a\rangle$, where $X_t = \left\\{\begin{matrix}-\nabla F_S(x) & \text{if } \Delta_t \leq 2\Delta_1 \newline 0 &\text{otherwise} \end{matrix}\right.$, and bound it by the Freedman’s inequality. Finally, by solving the inequality about the clipping parameter $\lambda$, we obtain a better parameter setting about step size $\eta$, thereby achieving a better convergence rate $\mathcal{O}(T^{\frac{2-2p}{3p-2}}\log^\frac{2p-2}{2p-1}\frac{T}{\delta})$, compared with $\mathcal{O}(T^{\frac{2-2p}{3p-2}}\log^2\frac{T}{\delta})$ in work [29].
>
> $\bullet$ **Direct extensions from [29].** Eq. (15) is a direct extension from [29]. Eq. (15) is a basic tool for handling heavy-tail noise. Many works [1, 2, 3] also depend on such results. Although our analysis depends on Eq. (15), there are many differences compared with [29]. The comprehensive comparisons are provided as following.
>
> [1] Nguyen. Improved convergence in high probability of clipped gradient methods with heavy tailed noise. NeurIPS.
>
> [2] Li. High probability analysis for non-convex stochastic optimization with clipping. ECAI 2023.
>
> [3] Cutkosky. High-probability bounds for non-convex stochastic optimization with heavy tails. NeurIPS, 2021.
>
> #### **3. Comparisons of theoretical results between our paper and [29].**
>
> (1) Different technologies. Under standard smoothness, [29] uses induction and constructs a martingale difference sequence
>
> $$U_s^t = \left\\{\begin{matrix} 0 & s = 0\newline \text{sgn} \left(\sum_{i=1}^{s-1} U_i^t\right) \frac{\langle \sum_{i=1}^{s-1}\beta^{t-i}\theta_i^a, \beta^{t-s}\theta_s^a\rangle}{\|\sum_{i=1}^{s-1}\beta^{t-i}\theta_i^a\|} & s\neq 0 \text{ and } \sum_{i=1}^{s-1}\beta^{t-i}\theta_i^a \neq 0\newline 0 & s\neq 0 \text{ and }  \sum_{i=1}^{s-1}\beta^{t-i}\theta_i^a = 0\end{matrix}\right.$$
>
> to handle the heavy-tailed noise, where $s \in \{0\} \cup [t]$ and $\beta = 1 - T^{-\frac{p}{3p-2}}$, while we construct a new martingale difference sequence $\sum_{t=0}^{l-1}(2\eta - L_0\eta^2 - L_1\eta^2\|\nabla F_S(x_t)\|)\langle X_t, \theta_t^a\rangle$, and solve a quadratic inequality about gradient $\|\nabla F_S(x_t )\|$ to handle the heavy-tailed noise.
>
> (2) Different intermediate results. As we propose new technologies mentioned above to deal with $(L_0, L_1)$-smoothness, our inductive Lemma 6 is very different from Lemma 8 in [29], which plays a key role in proving convergence rate. Specifically, due to different martingale difference sequences, each term in our Lemma 6 needs to be rebounded rather than simply extending the results in Lemma 8 in [29].
>
> (3) Respective advantages for convergence rates. Our goal is to achieve faster convergence rates under weaker assumptions. Our analysis achieves better convergence rate $\mathcal{O}(T^{\frac{2-2p}{3p-2}}\log^{\frac{2p-2}{2p-1}}\frac{T}{\delta})$ under the weaker $(L_0, L_1)$-smoothness as shown in Table 1. But [29] did not discuss this weaker assumption. On the contrary, under the stronger smoothness assumption, i.e., smoothness of individual stochastic function $f(x;\xi)$, [29] achieves a convergence rate $\mathcal{O}(T^{\frac{2-2p}{2p-1}}\log^2\frac{T}{\delta})$, which breaks the lower bound. But we did not discuss this stronger assumption.
>
> (4) Generalization. We prove the generalization bound but the work [29] did not discuss this issue. To the best of our knowledge, our paper is the first work that provides the generalization bound under the heavy-tailed noise and $(L_0, L_1)$-smooth setting.
>
> (5) Federated setting. We propose the novel FedCBG algorithm for heavy-tailed federated setting. More importantly, we propose novel error estimations as shown in Lemmas 1 and 2 to obtain the nearly optimal convergence rate and the first generalization bound. But [29] did not consider the federated case.
>
> To address your concerns, we will add a detailed discussion in our revised version.
>
> ### **About related works.**
>
> Thanks for your comments about related work. To address your concerns, we will add the following descriptions.
>
> As for federated learning, heavy-tailed noise also exists. The works [4, 5, 6] focus on this issue. [4] proposed a Adaptive Over-the-air Federated Learning (ADOTA-FL) algorithm for over-the-air model training. But it require both bounded gradient and bounded $p$-th moment for stochastic noise, which are stronger than our assumptions. [5] studied Byzantine resilience, communication efficiency and the optimal statistical error rates for edge federated learning under the heavy-tailed noise setting. For the non-convex analysis, in addition to requiring the boundedness of the gradient, it also requires the smoothness and second-order information of the stochastic gradient. The most related work to our paper is [6].
>
> [4] Wang C, Chen Z, Pappas N, et al. Adaptive federated learning over the air[J]. IEEE Transactions on Signal Processing, 2025.
>
> [5] Tao Y, Cui S, Xu W, et al. Byzantine-resilient federated learning at edge[J]. IEEE Transactions on Computers, 2023, 72(9): 2600-2614.
>
> [6] Yang H, Qiu P, Liu J. Taming fat-tailed (“heavier-tailed” with potentially infinite variance) noise in federated learning[J]. Advances in Neural Information Processing Systems, 2022, 35: 17017-17029.
>
> ### **For readability and some typos**
>
> $\bullet$  $\Delta_t = F_S(x_t) - F^*$. Thank you for pointing out the typos, we will revise them in our final paper.
>
> $\bullet$ We confirm that the bound in line 158 can be obtained by the Jensen's inequality or the Cauchy-Schwarz inequality. For the Jensen's inequality, $f\left(\frac{\sum_{t=1}^T a_t}{T}\right) \leq \frac{\sum_{t=1}^T f(a_t)}{T}$, we let $f(a) = a^2$ and $a_t = \|\nabla F_S(x_t)\|$, thus $\frac{1}{T}\sum_{t=1}^T\|\nabla F_S(x_t)\| = ((\frac{1}{T}\sum_{t=1}^T\|\nabla F_S(x_t)\|)^2)^{1/2} \leq (\frac{1}{T}\sum_{t=1}^T\|\nabla F_S(x_t)\|^2)^{1/2} \leq \mathcal{O}(\log^{\frac{p-1}{2p-1}}\frac{T}{\delta}/ T^{\frac{p-1}{3p-2}})$. For the Cauchy-Schwarz inequality, $\left(\sum_{t=1}^T a_t b_t\right)^2 \leq \left(\sum_{t=1}^T a_t^2\right)\left(\sum_{t=1}^T b_t^2\right)$, we let $a_t = \frac{1}{T}$ and $b_t = \|\nabla F_S(x_t)\|$ and can obtain the same result. Thus, both Jensen's inequality and Cauchy-Schwarz inequality can derive the result.
>
> $\bullet$ $\widetilde{\Delta}_t$ should be $\widetilde{\Delta}_t^i$. We modify the bracket error in line 294. Thank you very much for your detailed review.
>
> In summary, we will complete missing definitions, standardize the notation, and thoroughly proofread the manuscript in our revised version.

---

> > ### Comment · Reviewer_iqqA · 2025-08-07
> >
> > Thanks for the rebuttal. I will consider to raise the score after the reviewers' discussions.

---

> > > ### Author Response · Authors · 2025-08-08
> > >
> > > We sincerely appreciate your review and valuable questions that improved our work. If you have any further questions, please don't hesitate to reach out. We would be delighted to discuss them with you.

---

### Note · Authors · 2025-08-13

Dear reviewers, ACs, and SACs,

We would like to express our sincere gratitude for your valuable feedback, which has greatly contributed to improving our paper.

**In this paper, we prove tight high-probability bounds with proposed new analytical techniques for the Clipped-SGD algorithm and our proposed FedCBG algorithm under only weaker assumptions, i.e., $(L_0, L_1)$-smoothness and heavy-tailed noise conditions.**

In the centralized learning, we prove a faster convergence rate with high-probability for the clipped-SGD, which improves the existing high-probability bound. Our proposed analysis techniques are as follows. By induction, we bound the gradient norm explicitly by solving a quadratic inequality, construct new martingale difference sequences, and choose hyper-parameters by solving a new system of inequalities. Besides, we provide the first high-probability generalization error analysis without the gradient boundedness assumption.

In the federated setting, we propose a FedCBG algorithm under the two weaker assumptions. In addition to the scalable analytical techniques mentioned above, we prove pioneering analytical results for federated learning. We analyzed the batch gradient variance in Lemma 1 and proved the upper bound for stochastic errors in Lemma 2 under the heavy-tailed setting. Based on the two non-trivial techniques, FedCBG can achieve a nearly optimal high-probability convergence rate. Besides, we provide the first high-probability generalization error analysis for the federated setting.

Our submission received three reviews and we have dedicated considerable time and effort to addressing every concern and question raised. To provide clarity and thoroughly address all feedback, we have uploaded a detailed rebuttal and made careful revisions to the paper. We hope that our responses will assist you in gaining a comprehensive understanding of the paper, and we would be sincerely grateful for your recognition of our work.

Thank you once again for your thoughtful comments and positive discussions.

Sincerely,

The Authors

---

### Decision · Program_Chairs · 2025-09-17

**Decision:**

Accept (poster)

**Comment:**

The paper provides tight high probability convergence bounds for clipped SGD for non-convex optimization with heavy-tailed noise. The results hold for functions that are (L0, L1)-smooth, which is a generalization of the classical smoothness assumption. Previous works were either only in expectation, for a less general class of functions or smaller noise. The paper also has new bounds for the federated setting and especially generalization bounds, which are not common in the literature.

The reviewers generally agree that while many proof techniques are from previous works, combining them requires addressing several technical challenges. They all put the paper above the bar.